# The γ-tubulin meshwork assists in the recruitment of PCNA to chromatin in mammalian cells

Matthieu Corvaisier[1], Jingkai Zhou[1], Darina Malycheva[1], Nicola Cornella[1], Dimitrios Chioureas[2], Nina M. S. Gustafsson [2], Catalina Ana Rosselló [1], Silvia Ayora [3], Tongbin Li[4], Kristina Ekström-Holka[1], Karin Jirström[5], Lisa Lindström[1] & Maria Alvarado-Kristensson [1✉]

Changes in the location of γ-tubulin ensure cell survival and preserve genome integrity. We investigated whether the nuclear accumulation of γ-tubulin facilitates the transport of proliferating cell nuclear antigen (PCNA) between the cytosolic and the nuclear compartment in mammalian cells. We found that the γ-tubulin meshwork assists in the recruitment of PCNA to chromatin. Also, decreased levels of γ-tubulin reduce the nuclear pool of PCNA. In addition, the γ-tubulin C terminus encodes a PCNA-interacting peptide (PIP) motif, and a γ-tubulin–PIP-mutant affects the nuclear accumulation of PCNA. In a cell-free system, PCNA and γ-tubulin formed a complex. In tumors, there is a significant positive correlation between *TUBG1* and *PCNA* expression. Thus, we report a novel mechanism that constitutes the basis for tumor growth by which the γ-tubulin meshwork maintains indefinite proliferation by acting as an opportune scaffold for the transport of PCNA from the cytosol to the chromatin.

[1] Molecular Pathology, Department of Translational Medicine, Lund University, Skåne University Hospital Malmö, Malmö, Sweden. [2] Science for Life Laboratory, Department of Oncology and Pathology, Karolinska Institutet, Stockholm, Sweden. [3] Centro Nacional de Biotecnología (CNB-CSIC), Madrid, Spain. [4] AccuraScience LLC, Johnston, IA, USA. [5] Department of Clinical Sciences Lund, Oncology and Therapeutic Pathology, Lund University, Lund, Sweden. ✉email: maria.alvarado-kristensson@med.lu.se

A prerequisite for genome integrity is that cell division ends with one centrosome that orchestrates one end of the mitotic spindle to segregate one conserved genome into one newborn cell. To achieve this, cells have evolved surveillance mechanisms that prevent genomic instability by linking centrosome duplication with DNA replication and repair.

The protein γ-tubulin is an important regulator of both centrosome dynamics and of S phase progression[1–5]. Cytosolic γ-tubulin molecules combined with a variety of γ-tubulin complex protein (GCP)[6,7] form γ-tubulin ring complexes (γ-TuRCs), which nucleate microtubules[6–11] and regulate the mitotic spindle dynamics[12]. Part of the cellular pool of γ-tubulin also forms a complex with TCP-1 (CCT)[13], and that aggregate is involved in the correct folding of γ-tubulins in threads called γ-strings[11,14]. Nuclear envelope- and mitochondrial-associated γ-strings provide mitochondria and the nucleus with a structural scaffold that regulates the function of those organelles[11,15–19]. At G1/S transition, the phosphorylation of γ-tubulin regulates the recruitment of this protein to the centrosome and also promotes accumulation of γ-tubulin in the nucleus[2–4,20–24], where it enables centrosome replication and regulates the activities of E2 promoter-binding factors (E2Fs), respectively[1–4,23,25,26]. γ-Tubulin also accumulates in the nucleus in response to DNA double-strand breaks (DSBs)[3].

A coordinator of DNA replication and repair is the protein proliferating cell nuclear antigen (PCNA)[27,28]. PCNA is found in the cytosol during G1 phase and during progression through S phase accumulates in the nuclear compartment at specific sites (PCNA foci)[29–31]. In the chromatin, PCNA increases the efficiency of replicative polymerases[32–34]. PCNA expression is associated with advanced stages and with poor survival in gliomas and cervical cancer[27].

In this study, we assessed a possible function of γ-tubulin in facilitating the transport of cytosolic proteins into the nuclear compartment during DNA replication and neocarzinostatin (NCS)-mediated DSB repair. We describe how γ-tubulin and PCNA form a cellular complex, and this interaction facilitates the simultaneous accumulation of these two proteins in the nuclear compartment during both S phase entry and the NCS-induced DSB response. Our findings highlight an essential role for the γ-tubulin meshwork as a scaffold that assists in the transport of proteins between the cytosolic and the nuclear compartment in mammalian cells.

## Results

**Detection of chromatin-associated γ-tubulin.** Two *TUBG* genes and one pseudogene have been described in humans[35,36]. Although *TUBG1* is the predominantly expressed, *TUBG2* is expressed in the brain[10,37]. The protein sequences of γ-tubulin 1 (NP001061.2) and γ-tubulin 2 (BC009670.2) exhibits 97.55% homology, which has enabled a variety of commercially available antibodies that recognize both isoforms. To test the binding affinity of the antibodies, we tested four of them in cell lysates of human U2OS osteosarcoma cells and U2OS cells stably expressing *TUBG1* single guide (sg) RNA (green fluorescence protein [GFP]-tagged Cas9-CRISPR, knocks out *TUBG1* gene; *TUBG1*-sgRNA-U2OS)[11,38] and co-expressing either a sg-resistant *TUBG1* (*TUBG1*-sgRNA-U2OS-TUBG1) or a sg-resistant *TUBG2* (*TUBG1*-sgRNA-U2OS-TUBG2; Fig. 1a). Surprisingly, only one of the tested antibodies, T3320, displayed stronger immunolabeling of the ubiquitously expressed γ-tubulin 1 (Fig. 1a), whereas the rest of the antibodies recognized γ-tubulin 2. The lack of antibodies that recognized the ubiquitously expressed γ-tubulin 1 might have led to an underestimation of the levels of γ-tubulin protein in cells.

To evaluate the consequences of these observations, we used the four antibodies to evaluate various biochemically purified fractions of U2OS (express γ-tubulin 1), *TUBG1*-sgRNA-U2OS-TUBG1, and *TUBG1*-sgRNA-U2OS-TUBG2 cells, as well as human H9 neural stem cells (NSCs) that had differentiated into neurons (Fig. 1b, c; express both isoforms). Both γ-tubulin 1 and γ-tubulin 2 were found associated with chromatin, but only T3320 fully labeled it in U2OS cells (Fig. 1b). Furthermore, differentiation of NSCs into neurons decreased the expression levels of γ-tubulin 1, and once again, only T3320 fully recognized chromatin-associated γ-tubulin (Fig. 1c). Moreover, the antibody sc17788 labeled a protein band of the same size as γ-tubulin 2, and that band was found only in the cytoplasm (Fig. 1c). The observations demonstrate that detection of the chromatin-associated pool of γ-tubulin depends on two factors: the antibody used for detection and the degree of differentiation of the cell line.

Finally, to prove that the chromatin-associated pool of γ-tubulin did not represent a contamination from the cytosolic fraction caused by the presence of detergent in the lysis buffer, we pretreated cells with colcemid and cytochalasin B to depolymerize microtubules and actin filaments in cytoplasm and from nuclei[5,16,39]. Thereafter, we prepared the different fractions of the cells in the absence of detergent and then layered final fractions onto a sucrose gradient. We found γ-tubulin in the cytosolic fraction together with the chaperone TCP-1[13], and the microtubule component, α-tubulin, and also detected γ-tubulin in the chromatin fraction with histone (Fig. 1d). Furthermore, we found PCNA and γ-tubulin in the cytoplasm, nuclear membrane, and chromatin fractions, which confirmed the coexistence of those two proteins in the mentioned locations (Fig. 1d)[2,3,20,22,31,40]. Nonetheless, we were unable to detect components of the γ-TuRCs, such as GCP2, in the analyzed samples (Fig. 1d).

**PCNA and γ-tubulin accumulate simultaneously in chromatin.** Recent advances support the notion that γ-tubulin creates a cellular meshwork consisting of γ-strings (4–6 nm in diameter), γ-tubules (approximately 25 nm in diameter), and centrosomes[11,13,14,16]. In animal cells, γ-strings span from the cytosolic compartment through the membranes and into the chromatin (Fig. 2a)[11,13,14,16]. γ-Strings also nucleate on centrosomes and we found PCNA associated with γ-strings (Fig. 2a). During execution of S phase and in response to both NCS- or cisplatin-mediated DSB formation, γ-tubulin and PCNA accumulate simultaneously in the nucleus (Fig. 2b, c and Supplementary Fig. 1a, b)[1–4,32]. Consequently, we speculated that the association between PCNA and γ-strings and the simultaneous accumulation of both proteins in chromatin may reflect that the γ-tubulin meshwork facilitates the transport of cytosolic proteins, such as PCNA.

**Throughout cell division, γ-tubulin interacts with PCNA in the cytoplasm.** PCNA is a cytosolic protein (Fig. 1d)[31,40], which, due to its size (36 kDa), is predicted to passively diffuse into the nuclear compartment[29]. A nuclear localization sequence has been described that imports PCNA into the nucleus, where it forms distinct foci during execution of S phase[40]. Still, the mechanism underlying its nuclear accumulation and the factors that determine the size of the nuclear foci formed have not yet been fully elucidated. To identify functional links between PCNA and γ-tubulin, we investigated the amounts of PCNA associated with immunoprecipitates of γ-tubulin in various biochemical fractionations of NIH3T3 cells (Supplementary Fig. 2a) and of S phase-synchronized U2OS and MCF10A cells (Fig. 2d and Supplementary Fig. 2b, c). This revealed that PCNA and γ-tubulin co-immunoprecipitated throughout cell division and in all biochemical fractions from various cell types and species (Fig. 2d and Supplementary Fig. 2a–c). Furthermore, the

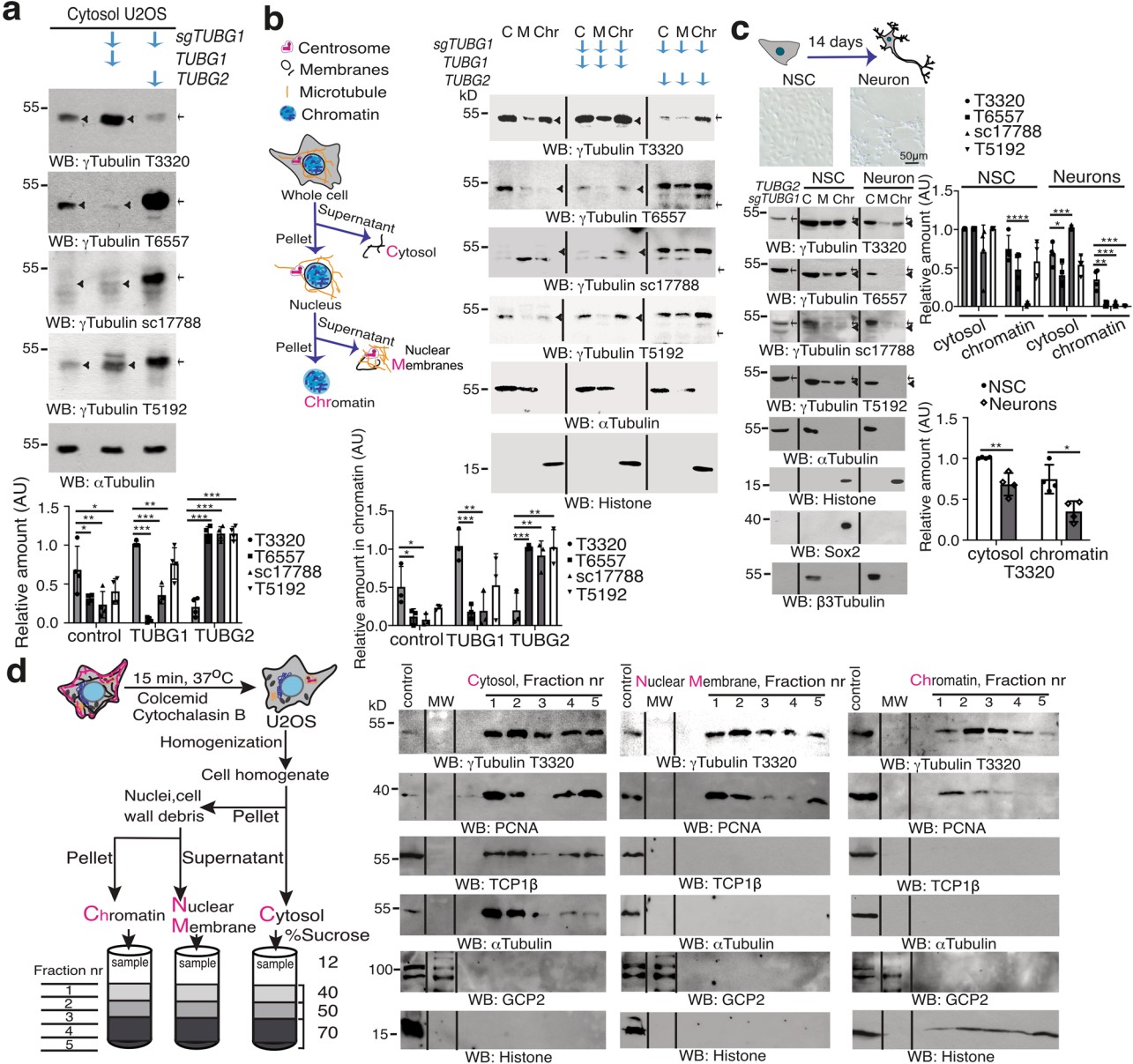

**Fig. 1 Despite 97.55% homology between γ-tubulin 1 and γ-tubulin 2, the proteins exhibit different conformations. a–c** The cytosolic fraction (**a**; $N = 4$) of U2OS cells or such cells stably expressing *TUBG1* single guide (sg) RNA (sg*TUBG1*, knocks out *TUBG1* gene) and co-expressing either a sg-resistant *TUBG1* or a sg-resistant *TUBG2* (**a**, **b**) was analyzed by western blotting (blots designated WB). **b**, **c** U2OS cells ($N = 3$), human H9 neural stem cells (NSCs), and H9 cells differentiated for 14 days into neurons ($1.0 \times 10^6$) were lysed, and proteins were biochemically divided into cytosolic (C), nuclear membrane (M), and chromatin fractions (Chr) and then analyzed by WB. **c** U2OS cells stably expressing *TUBG1* sgRNA and co-expressing a sg-resistant *TUBG2* (*TUBG2*sg*TUBG1*) were used as control ($N = 4$). **a–c** The expression of endogenous and recombinant γ-tubulin was analyzed using the following mouse (m) or rabbit (r) antibodies: T3320 (r) and sc17788 (m) raised against the C terminus of γ-tubulin; and T6557 (m) and T5192 (r) raised against the N terminus of γ-tubulin. Anti-Sox2 (NSC marker) and anti-β3-tubulin (neuronal marker) antibodies were used as markers of neuronal differentiation and anti-α-tubulin and anti-histone as loading controls. Black arrowheads and arrows indicate γ-tubulin 1 and γ-tubulin 2, respectively. **c** Microscopic images showing the cell morphology of whole NSCs and neurons, as indicated. The graphs in **a–c** illustrate densitometric analysis of the γ-tubulin content in the presented WB. To adjust for differences in protein loading, the concentration of a protein was determined as its ratio with α-tubulin (**a–c**) or histone (**b**, **c**). **d** In the absence of detergent, U2OS cells ($20 \times 10^6$) were biochemically divided into cytosolic, nuclear membrane, and chromatin fractions and then layered onto a sucrose gradient. The resulting fractions were analyzed by WB with the indicated antibodies. Total lysate from U2OS cells was run as control ($N = 5$). Molecular weight (MW) was loaded between samples and control. **b–d** Flow scheme of the biochemical fractionations performed (**b**, **d**) or of the morphological changes occurring in NSCs during differentiation (**c**). **b**, **d** The acronyms of the analyzed fractions are labeled in magenta. **a–d** Source data are provided in Supplementary Data 1.

γ-tubulin–PCNA complexes increased transiently in the chromatin during early S phase (Fig. 2d). Together, these findings indicate that the formation of PCNA–γ-tubulin complexes may be one mechanism that regulates the accumulation of PCNA in the nucleus.

**The nuclear accumulation of PCNA is dependent on γ-tubulin.** Inasmuch as PCNA and γ-tubulin interact in the cytoplasm throughout cell division, we speculated that, during G1 phase, the formation of cytosolic PCNA–γ-tubulin complexes may limit the transport of PCNA to the nuclear compartment. To test this

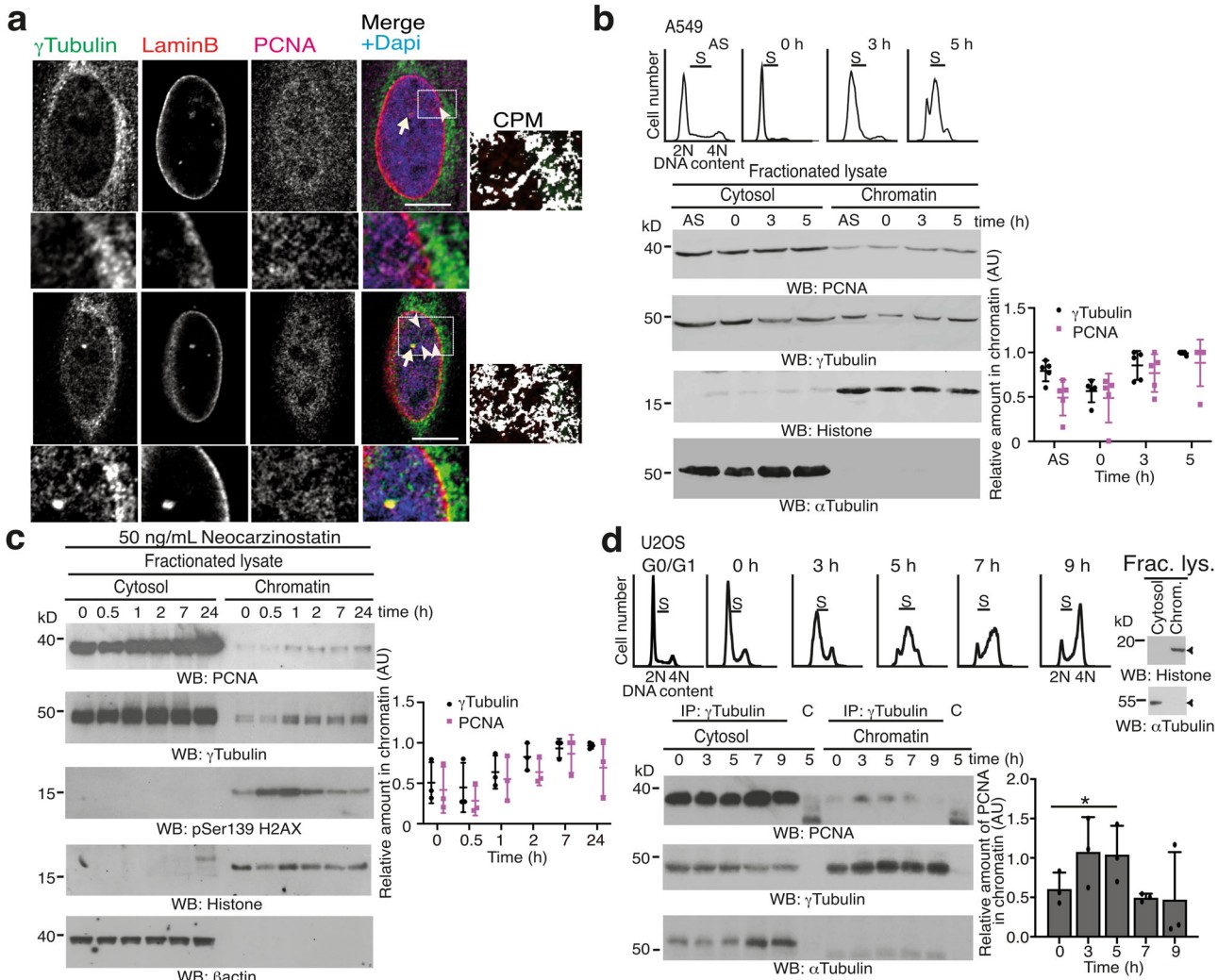

**Fig. 2 γ-Tubulin interacts with PCNA. a** Localization of γ-strings and PCNA in U2OS cells was examined by immunofluorescence staining with an anti-γ-tubulin, an anti-PCNA, and an anti-laminB1 (inner nuclear membrane protein) antibody. Nuclei were detected with DAPI. The confocal images show two different Z-planes from the nucleus of a cell. The white box indicates the magnified area shown in the insets. Scale bars: 10 μm (N = 4 samples). Colocalization pixel maps (CPM) represent the magenta/red and green channels of the magnified areas displayed in the inset. White areas denote colocalized pixels between channels. Arrowheads and arrows indicate γ-strings and centrosomes, respectively. **b** A549 cells (express γ-tubulin 1) were synchronized in early S phase by double thymidine block (0 h) and released for 3 h (early S) or 5 h (S). Cell cycle progression was monitored by determining the DNA content of cells with a nuclear counter (top panels). S phase synchronized and asynchronous (AS) A549 cells ($1.0 \times 10^6$) were lysed, and proteins were biochemically divided into cytosolic, and chromatin fractions and then analyzed by western blotting (WB) with antibodies against PCNA, γ-tubulin, α-tubulin, and histone (N = 5). **c** Biochemical fractions from neocarzinostatin-treated U2OS cells were examined by WB with the indicated antibodies against PCNA, γ-tubulin, phospho-histone2AX (marker of DSB), β-actin, and histone (N = 3; see also Supplementary Fig. 1). **d** Using extracts from S phase synchronized U2OS cells, γ-tubulin was immunoprecipitated with an anti-γ-tubulin or an isotype-matched control (C) antibody and developed by WB with the indicated antibodies (N = 3). The top panels illustrate the DNA content of the cells. Fractionated lysates (Frac. lys.) were run as control for the biochemical fractionation (right) and analyzed by WB using α-tubulin and histone as molecular markers for the cytosolic and nuclear fractions, respectively (see also Supplementary Fig. 2). The graphs in **b**–**d** illustrate densitometric analysis of the γ-tubulin and the PCNA chromatin content (**b**, **c**) or the content of PCNA in γ-tubulin immunoprecipitates from chromatin fractions in the presented WBs. To adjust for differences in protein loading, the concentration of a protein was determined as its ratio with histone (**b**, **c**) or with the immunoprecipitated protein (**d**) for each time point (N = 3). Source data are provided in Supplementary Data 1.

hypothesis, we used confocal microscopy to study the localization of the PCNA–γ-tubulin complexes in U2OS, *TUBG1*-sgRNA-U2OS, and *TUBG1*-sgRNA-U2OS-TUBG1 cells. We found that, in contrast to U2OS and *TUBG1*-sgRNA-U2OS-TUBG1 cells, the sg-mediated decreased levels of γ-tubulin in *TUBG1*-sgRNA-U2OS cells led to a reduced accumulation of PCNA and γ-tubulin in chromatin (Fig. 3a and Supplementary Fig. 3a, b). It should be noted that sg-mediated reduction of γ-tubulin expression also decreased the amount of γ-tubulin associated with centrosomes

but did not disturb the location of other centrosomal makers, such as pericentrin (Supplementary Fig. 3b). Nevertheless, the stable expression of *TUBG1* in *TUBG1*-sgRNA-U2OS cells restored the nuclear localization of PCNA, the formation of the PCNA–γ-tubulin complexes, and the association of γ-tubulin with centrosomes (Fig. 3a and Supplementary Fig. 3a, b). Moreover, expression of *TUBG1*-sgRNA prevented the immunofluorescence staining of chromatin-associated PCNA in an S phase-synchronized U2OS cell population (Fig. 3b). But *PCNA*

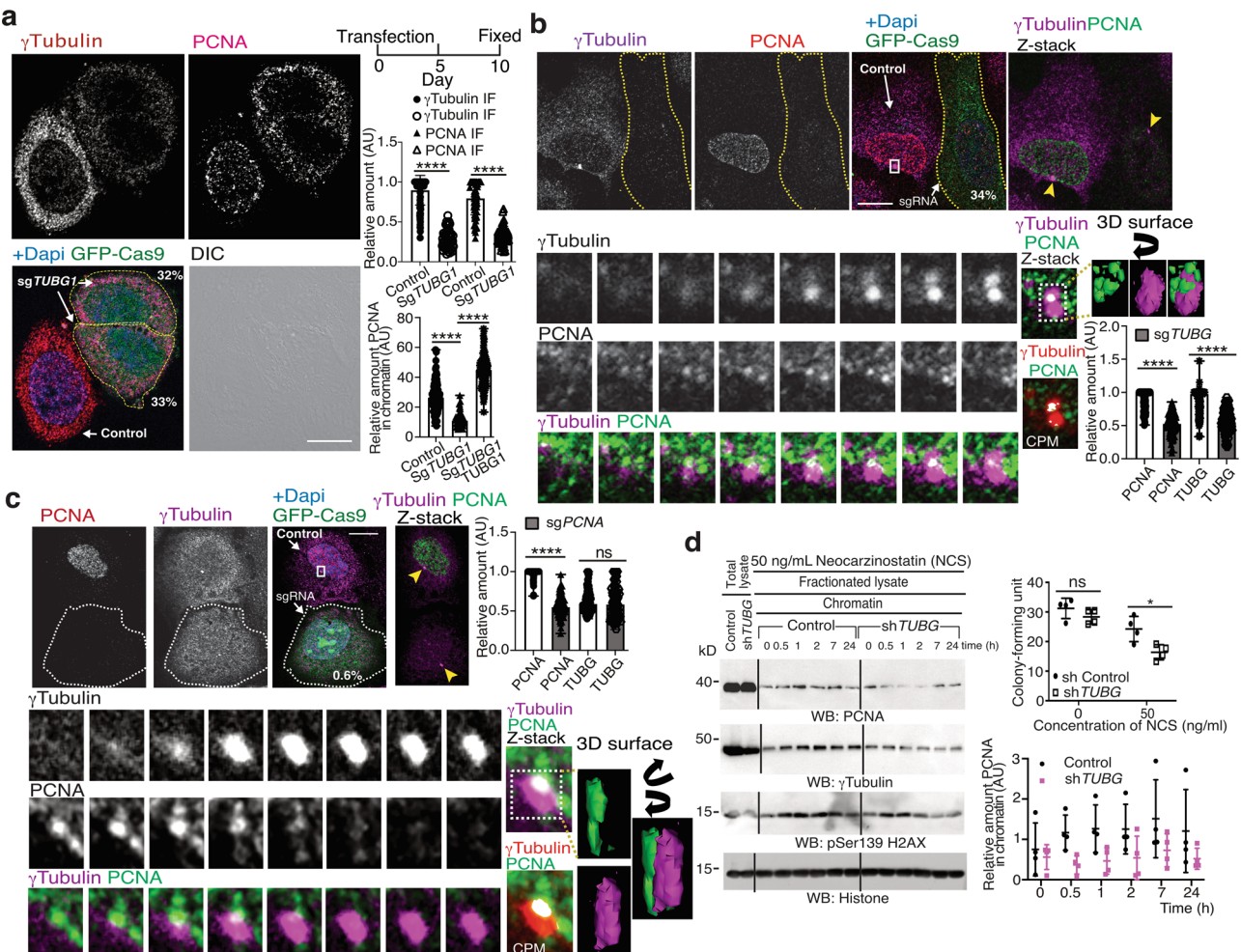

**Fig. 3 The nuclear levels of γ-tubulin regulate the recruitment of PCNA to the nuclear compartment. a–c** The differential interference contrast (DIC)/ fluorescence (**a**) images show U2OS cells expressing human *TUBG*-sgRNA (**a**, **b**) or *PCNA*-sgRNA (GFP-Cas9) for 10 days (**c**) that were synchronized in early S phase by double thymidine block and then released for 5 h (**b**, **c**). Localization of endogenous γ-tubulin and PCNA was examined by immunofluorescence staining with the indicated antibodies, and nuclei were detected with DAPI in U2OS or in U2OS cells expressing the following constructs: *TUBG*-sgRNA (**a**, **b**), *PCNA*-sgRNA (**c**), and co-expressing *TUBG*-sgRNA and a sg-resistant *TUBG1* (**a**). Scale bars: 10 μm (*N* = 6). Z-stacks show average intensity projection of sequential images (**b** 14, **c** 13) that were collected at 0.2-μm intervals. The cropped image stack of the centrosome and the interacting PCNA foci were visualized in a 3D surface image with the 3D viewer plugin of ImageJ[70]. The 3D surface image obtained was rotated according to the direction of the black arrows. **a** Within samples, quantification of γ-tubulin and of the PCNA signal in chromatin was done with the ImageJ software by comparison of immunofluorescence-labeled γ-tubulin and PCNA either in asynchronous (**a**) or in synchronous (**b**, **c**) cells expressing Cas9-crispGFP (dashed lines; **a** sg*TUBG1*, *N* = 52 cells; **b** sg*TUBG1*, *N* = 100 cells; **c** sg*PCNA*, *N* = 102 cells) with non-expressing cells (**a** control, *N* = 100 cells; **b** control, *N* = 100 cells; **c** control, *N* = 102 cells) or with cells co-expressing *TUBG*-sgRNA and a sg-resistant *TUBG1* (addback, *N* = 101 cells). The graphs illustrate the mean value of γ-tubulin and of the PCNA signal in chromatin (mean ± SD; ****P < 0.0001; see also Supplementary Fig. 3). **a**, **b** Numbers in images indicate the remaining protein levels of γ-tubulin or PCNA relative to control in the indicated cells. **b**, **c** White boxes indicate the magnified area shown in the insets. Sequential inset images illustrate the interaction of the centrosome with one of the PCNA foci. Colocalization pixel maps (CPM) represent the magenta/ red and green channels of the magnified areas displayed in the inset. White areas denote colocalized pixels between channels. Arrowheads indicate centrosomes. **d** Control U2OS cells or U2OS cells stably expressing *TUBG* short hairpin RNA (shRNA) were treated and biochemically fractionated as in Fig. 2b and analyzed by western blotting (WB) with the indicated antibodies (see also Supplementary Fig. 1). The graphs illustrate the colony-forming unit of the cell populations after NCS treatment (mean ± SD; *N* = 3, *P < 0.05; top) and the relative densitometric analysis of the γ-tubulin and the PCNA chromatin content in the illustrated WBs (mean ± SD; *N* = 4, **P < 0.01; bottom). To adjust for differences in protein loading, the concentration of a protein was determined as its ratio with histone for each time point (see also Supplementary Fig. 4). **a**, **d** Source data are provided in Supplementary Data 1.

sgRNA expression did not block the immunofluorescence staining of nuclear γ-tubulin (Fig. 3c), suggesting a γ-tubulin-dependent location of PCNA in the chromatin.

**Reduced levels of γ-tubulin decrease the chromatin loading of PCNA.** Due to the cytotoxicity of sgRNA-induced reduction of either γ-tubulin or PCNA[11,38,41], we studied the effects of

reducing the levels of γ-tubulin during the recruitment of PCNA by stably expressing *TUBG* short hairpin RNA (shRNA) in U2OS cells (*TUBG*-shRNA-U2OS). The expression of *TUBG* shRNA decreased the endogenous γ-tubulin pool by ~40–50% (Fig. 3d)[11]. Analysis of *TUBG*-shRNA-U2OS cells showed that in comparison with U2OS cells (control) both the chromatin loading of PCNA and the colony-forming ability were lowered by treatment with NCS (Fig. 3d and Supplementary Fig. 4). These observations

further confirm that the γ-tubulin meshwork is involved in the loading of PCNA to chromatin, and they also indicate that the nuclear accumulation of γ-tubulin upon DSBs is necessary for the survival of the cells.

To investigate possible colocalization of PCNA and γ-tubulin in the chromatin, we synchronized cells in early S phase (Fig. 4a) and analyzed whether γ-tubulin immunoprecipitates specific chromatin sites during cell division by sequencing the chromatin immunoprecipitates of γ-tubulin (chromatin immunoprecipitation–sequencing (ChIP-seq)) from U2OS cell populations. Peak calling was performed using PeakSeq (version 1.31)[11]. As cells entered S phase, γ-tubulin accumulated in the chromatin (Fig. 2b)[1–3]. Accordingly, we found an increase in the number of the peaks of genomic DNA associated with γ-tubulin as the cells progressed through S phase (3–7 h; Fig. 4a). ChIP-seq analysis of PCNA and macro histone 2A1 immunoprecipitates from S phase-synchronized U2OS cell populations (3 and 5 h; Fig. 4a) revealed that only the PCNA peaks showed a clear propensity to occur in close proximity to the γ-tubulin peaks; this propensity was not found between macro histone 2A1 and γ-tubulin peaks (Fig. 4a), supporting an interdependence in the chromatin localization of PCNA and γ-tubulin.

As the next step, we mapped the location of γ-tubulin and PCNA on the genomic DNA of MCF10A and of MCF10A cell populations that stably expressed TUBG shRNA (TUBG-shRNA-MCF10A) by performing ChIP-seq analysis of γ-tubulin and PCNA immunoprecipitates (Fig. 4b and Supplementary Fig. 5a). To avoid possible artefacts caused by the method used for the assessment, peaks were called with MACS2 (version 2.1.1)[11]. The identified peaks associated with nuclear chromatin were more numerous in γ-tubulin immunoprecipitates from MCF10A cells than in such precipitates from TUBG-shRNA-MCF10A cells, proving that shRNAi-mediated reduction of γ-tubulin levels led to a decrease in binding of γ-tubulin to genomic DNA (Fig. 4b and Supplementary Fig. 5a).

Similarly, the number of PCNA peaks increased as the cells progressed through S phase (2 h; Fig. 4b and Supplementary Fig. 5b). The identified differential peaks associated with DNA at 1 and 2 h were more abundant in PCNA immunoprecipitates from MCF10A cells than in such precipitates from TUBG-shRNA-MCF10A cells. This indicated that shRNAi-mediated reduction of γ-tubulin levels led to a reduced loading of PCNA on chromatin (Fig. 4b and Supplementary Fig. 5b), which is further evidence that γ-tubulin is involved in the association of cytosolic PCNA with chromatin.

**PCNA and γ-tubulin coincide at DNA sequence motifs enriched in replication origins**. To determine the number of PCNA and γ-tubulin peaks that overlapped, we defined that two peaks overlapped if at least 25% of the span of the γ-tubulin and PCNA peak sequence overlapped. We found that as MCF10A cells progressed through S phase, 460 (1 h; ~82% of PCNA peaks) and 1461 (2 h; ~95% of γ-tubulin peaks) of the 1035 (1 h) and 1531 (2 h) γ-tubulin peaks overlapped with at least one of the 562 (1 h) and the 2745 (2 h) PCNA peaks (Fig. 4b). Similar results were obtained regarding the peaks found in immunoprecipitates from S phase-synchronized TUBG-shRNA-MCF10A cells. We also noted that, of the 288 (1 h) and 79 (2 h) γ-tubulin peaks, 143 (1 h; ~50% of γ-tubulin peaks) and 38 (2 h; ~48% of γ-tubulin peaks) overlapped with at least one of the 290 (1 h) and the 399 (2 h) PCNA peaks. We subsequently randomized the widths of the locations of the PCNA peaks 2000 times and found that in none of the 2000 times instances did the number of overlapped peaks exceed the previously observed number of overlapped peaks even once (the P value was <0.005 in all cases). These results prove that PCNA and γ-tubulin are frequently localized to the same DNA region.

We next sought to investigate the enriched motifs in the sequences occupied by γ-tubulin that overlapped with PCNA. We found that, early in S phase (1 h), among the 460 γ-tubulin peaks found in MCF10A cells that overlapped with PCNA peaks, the most significantly enriched motif ($e = 7.9 \times 10^{-2161}$) was the purine-rich element binding protein A (PURA) humanH10Mo.D motif, which was observed at origins of replication and in gene flanking regions (Fig. 4c)[42–44]. The PURA motif was present in 181 γ-tubulin peaks, 178 of which overlapped with PCNA peaks. To confirm that γ-tubulin localized to DNA regions containing the PURA motif, we performed a ChIP assay using γ-tubulin antibodies in U2OS cells and in U2OS cell populations that stably expressed TUBG shRNA (TUBG-shRNA-U2OS) or that stably co-expressed TUBG shRNA and a C-tagged TUBG1-GFP shRNA-resistant gene (TUBG-shRNA-U2OS-TUBG1-GFP; compensates the TUBG shRNA-mediated reduction of γ-tubulin). The PURA motif was present in higher amounts in γ-tubulin immunoprecipitates from U2OS cells than in such precipitates from TUBG-shRNA-U2OS cells (Fig. 4d and Supplementary Fig. 5). Expression of GFP-γ-tubulin in TUBG-shRNA-U2OS cells compensated this reduction and confirmed that modulation of the expression levels of γ-tubulin affect the recovery of the PURA motif (Fig. 4d and Supplementary Fig. 5), strongly suggesting that γ-tubulin frequently localizes at origins of replication together with PCNA[42–44].

**γ-Tubulin is enriched at active origins of replication**. As cells exit mitosis, origin recognition complex protein (Orc) binds to replication origins. Thereafter, the replicative helicase, Mcm, together with additional proteins are recruited, forming the pre-replication complex that licenses DNA replication[45]. In S phase, most of the origins of replication remain dormant, and only in a subset of origins, the replicative helicase of Mcms is activated, and replication factor C complex (RFC) loads PCNA, which characterized active origins of replication[34,46]. To explore the contribution of γ-tubulin to origin activation, we synchronized U2OS and TUBG-shRNA-U2OS cell populations and mapped the location of γ-tubulin, PCNA, Mcm5, and the transcription factor FoxM1 on the genomic DNA by performing ChIP-seq analysis of the resulting immunoprecipitates (Fig. 4e). Peak calling was performed with MACS2 (version 2.1.1)[11]. We considered Mcm5 peaks overlapping with PCNA peaks (by at least 25% for both peaks) as active origins and those not overlapping with PCNA as dormant. We found that, across all samples, the overlapping between γ-tubulin peaks and the identified active and dormant origins of replication was 96.8 and 51%, respectively, altogether confirming the presence of γ-tubulin in almost all the identified active origin of replication and in half of the dormant domains (Fig. 4e and Supplementary Table 1). In contrast, the overlapping between FoxM1 peaks and the identified active and dormant origins of replication was 74.7 and 31.1%, respectively, which confirms that firing origins are enriched close to transcription start sites (Fig. 4e and Supplementary Table 1)[47]. Analysis of γ-tubulin and PCNA immunoprecipitates from TUBG-shRNA-U2OS cells showed that in comparison with U2OS cells (control) there was a 48% reduction of the number of γ-tubulin peaks across all samples (Fig. 4e). But the reduction in the number of PCNA peaks precipitated varied over time. We found that, similarly to MCF10A, the shRNAi-mediated reduction of γ-tubulin levels led to a 52% reduction in the number of identified differential peaks associated with DNA that was retrieved from PCNA immunoprecipitates from early S phase synchronous TUBG-shRNA-U2OS cell populations (3 h, Fig. 4e). As cell division progressed (7, 9 h), the number of retrieved peaks from PCNA precipitates was lower in U2OS cell populations than from

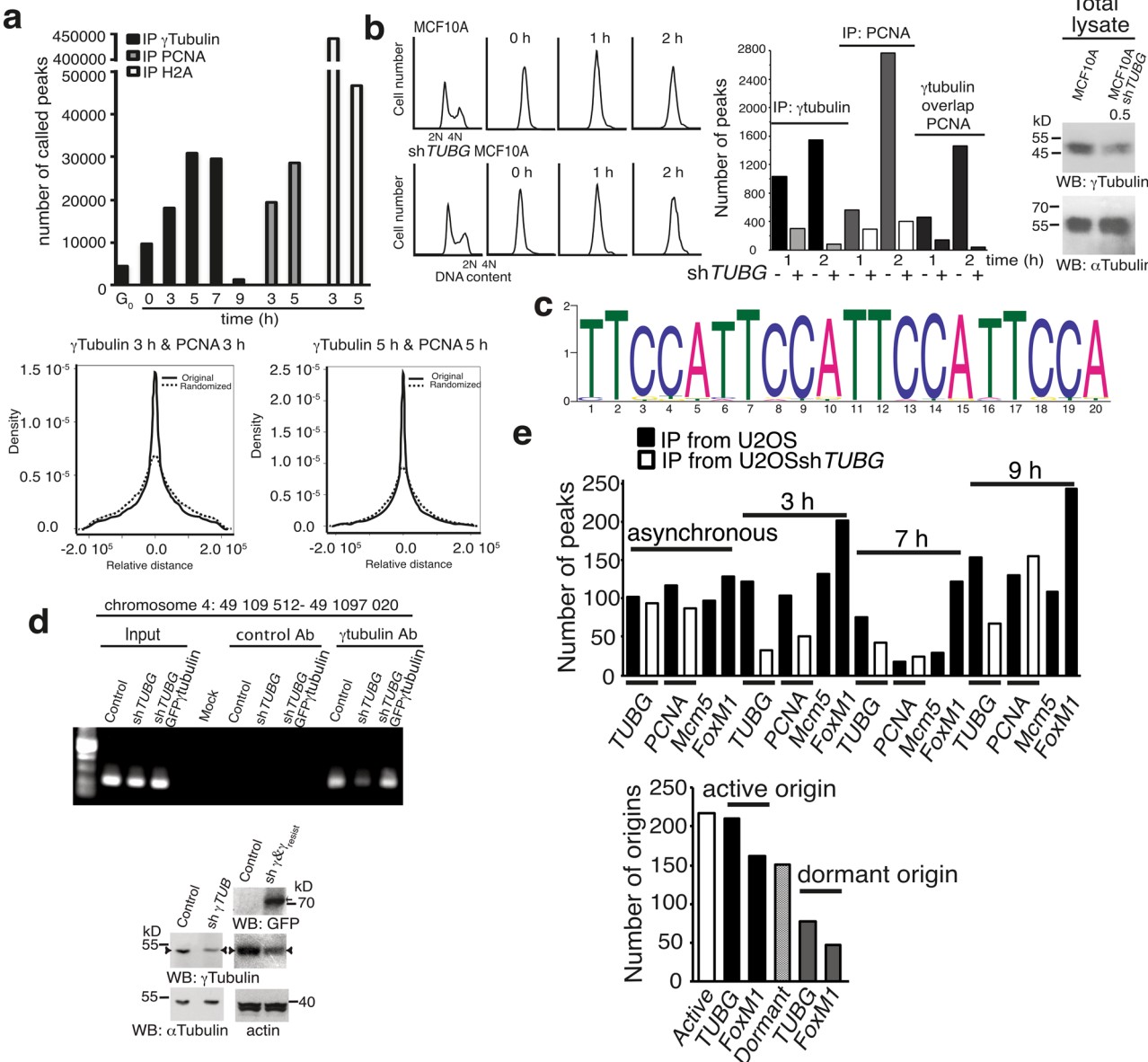

**Fig. 4 PCNA and γ-tubulin bind to chromatin on the same DNA-binding motif. a** U2OS cells were synchronized in G₀/G₁ or in early S phase (0 h) and then released for 3 and 5 h (S) or 7 (S–G₂/M) and 9 h (G₂/M). To identify the locations where γ-tubulin, PCNA, and macro histone 2A (H2A) accumulate in the chromatin, we sequenced the DNA-associated with chromatin immunoprecipitates from these proteins (γ-tubulin: G₀, 0, 3, 5, 7, and 9 h; PCNA and H2A: 3 and 5 h). The graphs show the number of binding sites (peaks called; top) found in the human genome to which an indicated protein binds or the distance distributions between a γ-tubulin peak and the PCNA peak closest to it (bottom). **b** MCF10A and MCF10A cells stably expressing *TUBG* shRNA (MCF10Ash*TUBG*) were synchronized in early S phase (0 h) and released for 1 and 2 h. Cell cycle progression was monitored by determining the DNA content of cells (graphs; N = 3). Total lysates from MCF10A and MCF10Ash*TUBG* were analyzed by WB for the expression of endogenous γ-tubulin and α-tubulin (loading control; N = 3). The numbers above the blots (WB) indicate the level of depletion of γ-tubulin relative to control. To adjust for differences in protein loading, the concentration of γ-tubulin was determined as its ratio with α-tubulin. To map the DNA region where γ-tubulin accumulates in the chromatin, we sequenced the DNA associated with chromatin immunoprecipitates from γ-tubulin and PCNA. Graphs show the number of peaks called in the human genome where γ-tubulin or PCNA accumulates or for which γ-tubulin and PCNA peaks overlapped at the indicated times (see also Supplementary Fig. 5). **c** Scheme of the significantly enriched purine-rich element binding protein A (PURA) DNA-binding motif found in the chromatin-associated with immunoprecipitates of γ-tubulin that overlapped with PCNA at 1 h. **d** U2OS, U2OSsh*TUBG*, and U2OSsh*TUBG* cells expressing a sh*TUBG*-resistant GFP-γ-tubulin1 were analyzed by ChIP using an anti-γ-tubulin antibody. PCR primers amplified the indicated region (N = 3). **e** The DNA-associated with chromatin immunoprecipitates from γ-tubulin, PCNA, Mcm5, and FoxM1 from asynchronous and synchronized U2OS and U2OSsh*TUBG* cells as in Fig. 4a and then released for 3 (S) and 7 h (S–G₂/M) or 9 h (G₂/M) was sequenced. Active origins are Mcm5 peaks that overlap with PCNA peaks, and dormant are those Mcm5 peaks that do not overlap with PCNA. The graphs show the number of peaks called in the human genome to which an indicated protein binds at the indicated time (top) or the number of origins of replication where γ-tubulin and FoxM1 were found (bottom), respectively (N = 2; see also Supplementary Table 1). **b**, **d** Source data are provided in Supplementary Data 1.

*TUBG*-shRNA-U2OS cells (Fig. 4e). These data reflects the previously reported delay in S phase progression in cells with lowered expression of γ-tubulin[2,11], which either activates compensatory mechanisms to overcome the reduced γ-tubulin levels or triggers accumulation of PCNA at fired stalled replication origins.

**γ-Tubulin coincides with PCNA in distinct nuclear foci.** To monitor the effect of the γ-tubulin meshwork in the nuclear accumulation of PCNA, we analyzed Z-stack images of whole fixed cells. Immunofluorescence assessment of endogenous γ-tubulin in S phase-synchronized U2OS cell populations revealed the accumulation of γ-tubulin and PCNA in distinct chromatin foci (Fig. 5a)[48]. γ-Strings connected the γ-tubulin and PCNA foci with the γ-tubulin in the pericentriolar matrix (PCM) and once again we found PCNA associated with those γ-strings (Fig. 5a). Also, PCNA foci became associated with γ-tubulin in γ-strings and in the PCM (Fig. 3b, c and Supplementary Fig. 3a), further implying that centrosome-associated γ-strings may assist in the transport of PCNA to the nucleus to form chromatin-associated PCNA foci.

**The dynamics of γ-strings, γ-tubules, centrosomes, and PCNA during S phase.** Inasmuch as the PCM moves around the nuclear envelope[5], we visualized these movements during the nuclear accumulation of endogenous PCNA in living *TUBG*-shRNA-U2OS-TUBG1-GFP cells by recording Z-stack time-lapse images of whole *TUBG*-shRNA-U2OS-TUBG1-GFP cells stably expressing the red fluorescence protein (RFP)-fused V$_H$H anti-PCNA antibody called cell cycle chromobody (CCC; *TUBG*-shRNA-U2OS-TUBG1-CCC)[49]. As the CCC foci were formed, the PCM was in contact with various expanding CCC complexes (Fig. 5b and Supplementary Movie 1), strongly suggesting that the γ-strings associated with the PCM assist in the transport of cytosolic PCNA to the growing nuclear CCC foci.

Further analysis of time-lapse images from *TUBG*-shRNA-U2OS-TUBG1-CCC cells revealed that some of the γ-strings in the nucleus were organized in various γ-strings-rich nuclear bodies (γSNBs)[48]. Differential interference contrast (DIC) images showed that the γSNBs were localized in defined nuclear domains that in fixed cells were stained with the nucleolar marker nucleolin (Fig. 5c and Supplementary Fig. 6)[22]. As S phase progressed, CCC nucleated around γSNBs to form large CCC complexes (Fig. 5c and Supplementary Movie 2), whereas γ-tubules in the cytosol were transiently formed close to the PCM and/or nuclear envelope (380', 415', 470', 475', 495', and 505'; Fig. 5c and Supplementary Movie 2)[48]. Moreover, γ-strings connected γSNBs with each other and with the centrosome and nuclear envelope (380'–415', and 465'–570'; Fig. 5c and Supplementary Movie 2). Immunofluorescence staining with an anti-γ-tubulin antibody confirmed the presence of GFP-positive γ-tubules in the monitored cells (Supplementary Fig. 7). These findings suggest that the γ-tubulin meshwork acts as a supporting scaffold during the recruitment of PCNA to the chromatin.

**Mutation of the PCNA-interacting peptide motif on γ-tubulin attenuates the nuclear accumulation of PCNA.** To define what region of γ-tubulin interacts with PCNA, we used two fragments of γ-tubulin fused to GFP: N-γ-tubulin$^{1-335}$ and C-γ-tubulin$^{336-451}$. Immunoprecipitation of GFP from U2OS and *TUBG*-shRNA-U2OS cells stably expressing GFP-γ-tubulin, GFP-N-γ-tubulin$^{1-335}$, or GFP-C-γ-tubulin$^{336-451}$ showed that both GFP-γ-tubulin and GFP-C-γ-tubulin$^{336-451}$ exhibited a stronger binding to PCNA than GFP-N-γ-tubulin$^{1-335}$ did (Fig. 6a and supplementary Fig. 8a), which suggest that the C terminus of γ-

tubulin interacts directly with PCNA. We noted that the C-terminal region of γ-tubulin included a putative PCNA-interacting peptide (PIP) motif (QXX[M/L/I]XX[F/Y][F/Y], established based on the proteins known to interact with PCNA[50]) comprising residues Q429, I432, and Y435 (Fig. 6b). Sequence alignments showed that, considering all the tubulins in animal species, the residues in the PIP motif are conserved only in *TUBG1* and *TUBG2* (Fig. 6b)[51].

To confirm both a direct interaction between PCNA and γ-tubulin and that mutation of the PIP motif disrupts the formation of the complex in a cell-free system, we used a cell-free system to determine whether bacterially produced His-PCNA would interact with any of the following: a GST-tagged γ-tubulin; GST-γ-tubulin containing the mutations Q429A, I432A, and Y435A (γ-tubulin$^{A429-A432-A435}$); or the control recombinant protein GST-CDC25C$^{200-256}$ (Fig. 6c). In experiments in vitro, His-PCNA interacted with GST-γ-tubulin but not with GST-CDC25C$^{200-256}$ (Fig. 6c). We detected a significantly decreased amount of PCNA–γ-tubulin complexes in samples containing γ-tubulin$^{A429-A432-A435}$ (**$p < 0.01$; Fig. 6c), demonstrating that the γ-tubulin residues Q429, I432, and Y435 assist in the formation of the PCNA–γ-tubulin complexes.

Next, we investigated the effect of disrupting PCNA interaction with γ-tubulin$^{A429-A432-A435}$ in vivo by stably depleting the endogenous γ-tubulin pool with sgRNA (*TUBG1*-sgRNA-U2OS)[11,38] and co-expressing either a sg-resistant *TUBG1* (*TUBG1*-sgRNA-U2OS-TUBG1) or a sg-resistant *TUBG1*$^{A429-A432-A435}$ (*TUBG1*-sgRNA-U2OS-TUBG1$^{A429-A432-A435}$; Supplementary Fig. 8b). These stable cell lines expressed similar amounts of cytosolic PCNA but higher protein levels of γ-tubulin$^{A429-A432-A435}$ (Supplementary Fig. 8b). Note that amino acid changes in the mutated region caused a size shift in sodium dodecyl sulfate gels (Supplementary Fig. 8b), as previously reported[10]. Additional characterization showed that levels of PCNA were higher in immunoprecipitates from γ-tubulin than from γ-tubulin$^{A429-A432-A435}$ (Fig. 6d), demonstrating that the PIP motif present on the C terminus of γ-tubulin mediates the interaction with PCNA.

Thereafter, we studied the impact of γ-tubulin$^{wildtype}$ and γ-tubulin$^{A429-A432-A435}$ on the immediate recruitment of PCNA upon NCS-mediated DSB formation. NCS treatment of *TUBG1*-sgRNA-U2OS-TUBG1 and *TUBG1*-sgRNA-U2OS-TUBG1$^{A429-A432-A435}$ cells triggered a larger enrichment of γ-tubulin$^{A429-A432-A435}$ in chromatin in comparison with the enrichment found of γ-tubulin$^{wildtype}$ (Fig. 6e). In contrast, the early accumulation of PCNA in the chromatin fraction after 1 h of treatment with NCS was impaired only in *TUBG1*-sgRNA-U2OS-TUBG1$^{A429-A432-A435}$ cells (Fig. 6e). These findings demonstrate that the PIP motif is the motif that may assist in the rapid transport of cytosolic PCNA to the chromatin.

**The γ-tubulin PIP motif regulates cell cycle progression.** To study the effect of mutation of the PIP motif on the recruitment of PCNA to the chromatin during cell cycle progression in living cells, we stably co-expressed CCC in *TUBG1*-sgRNA-U2OS-TUBG1 (*TUBG1*-sgRNA-U2OS-TUBG1-CCC) and *TUBG1*-sgRNA-U2OS-TUBG1$^{A429-A432-A435}$ (*TUBG1*-sgRNA-U2OS-TUBG1$^{A429-A432-A435}$-CCC) and monitored cell cycle progression by time-lapse imaging. In comparison with *TUBG1*-sgRNA-U2OS-TUBG1-CCC cells (Fig. 7a), CCC-expressing *TUBG1*-sgRNA-U2OS-TUBG1$^{A429-A432-A435}$ cells showed a reduction in the intensity of CCC complexes in the nucleus and nucleoli during S phase progression (Fig. 7a–c). These data further suggest that mutation of the PIP domain affects the interaction of γ-tubulin with PCNA,

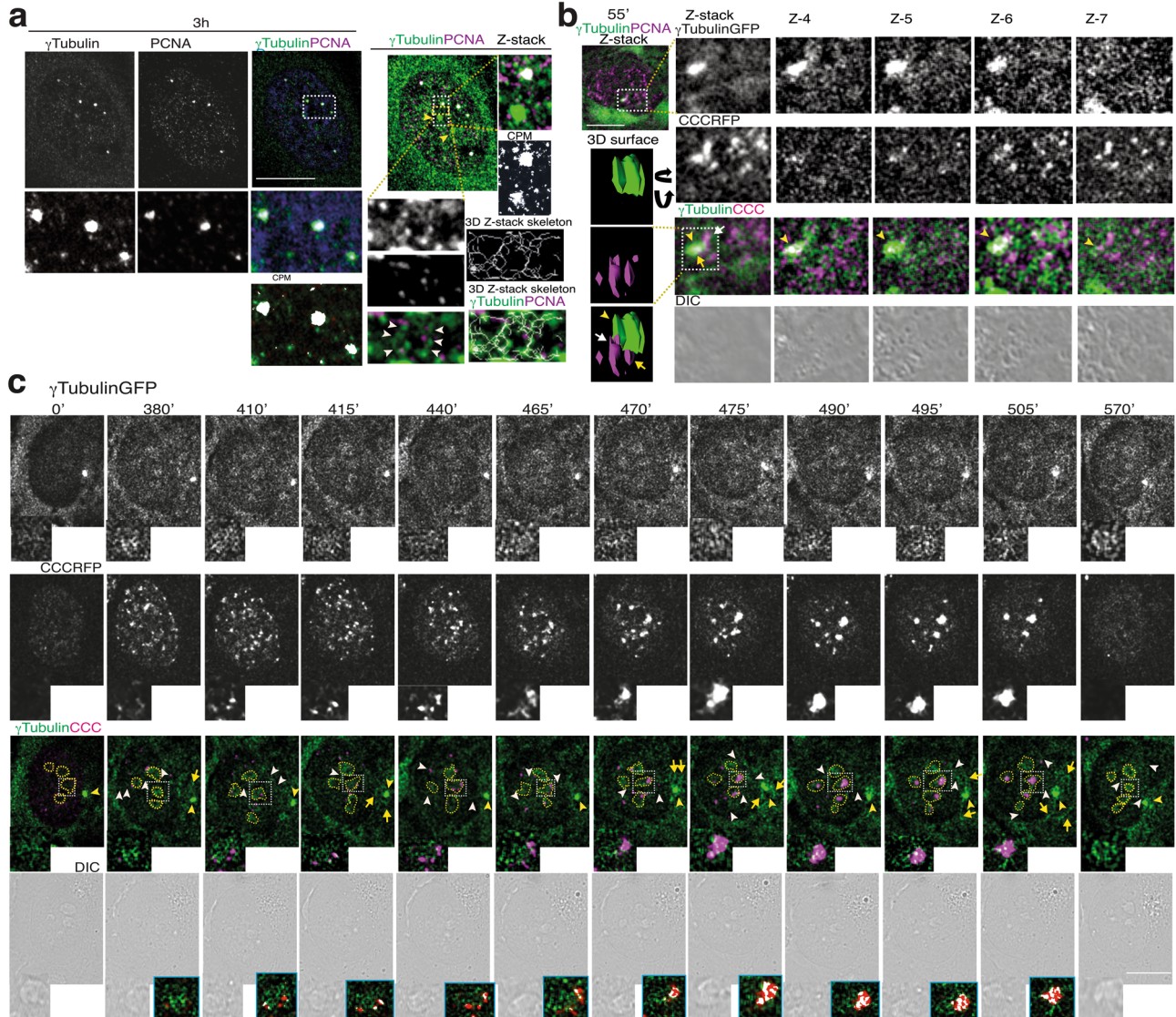

**Fig. 5 γ-Tubulin and PCNA accumulates at γ-strings-rich nuclear bodies. a** U2OS cells were synchronized in early S phase and then released for 3 h. Localization of endogenous γ-tubulin and PCNA was examined by immunofluorescence staining with the indicated antibodies, and nuclei were detected with DAPI. The confocal image shows one plane of the nucleus. **a**, **b** Z-stacks show average intensity projection of sequential images (**a** 13, **b** 10) that were collected at 0.2-μm (**a**) or 0.4-μm (**b**) intervals. **a** A 3D thinning algorithm was applied to the Z-stack images from the green channel to find the centerlines (skeleton) of structures in the input images[69]. **b**, **c** The differential interference contrast (DIC)/fluorescence images present one lapse of a time-lapse Z-stack series (**b**) or time-lapse series (**c**) of a stable *TUBG*sh-U2OS cell co-expressing GFP-γ-tubulin$_{resist}$ (γTubulinGFP) and cell cycle chromobody (CCCRFP, to visualize the nuclear location of endogenous PCNA in a living cell). **b** The image series show chosen cropped Z-frames of the interaction of two of the CCC foci with the PCM. The cropped image stack of the centrosome (yellow arrowhead) and the interacting CCC foci (white and yellow arrows) were visualized in a 3D surface image with the 3D viewer plugin of ImageJ[70]. The 3D surface image obtained was rotated according to the direction of the black arrows. **c** The image series show chosen frames of the location of PCNA and illustrate how the γ-tubulin meshwork assists in the recruitment of PCNA (N = 30 cells). One of the series shows γ-strings-rich nuclear bodies (yellow dashed lines), with magnified areas in the insets (white dashed boxes) depicting the recruitment of PCNA to a nucleolus (see also Supplementary Figs. 6 and 7). **a**–**c** The white boxes indicate the magnified area shown in the insets. Yellow and white arrowheads point out centrosomes and γ-strings between various cellular structures, respectively (N = 80), and yellow arrows indicate γ-tubules. Colocalization pixel maps (CPM or blue boxes) represent the magenta/red and green channels of the magnified areas displayed in the inset. White areas denote colocalized pixels between channels. Scale bars: 10 μm.

resulting in both a reduced accumulation of CCC in chromatin and the formation of smaller CCC foci (Fig. 7, Supplementary Fig. 9, and Supplementary Movies 3 and 4). Accordingly, the colony-forming ability of asynchronous *TUBG1*-sgRNA-U2OS-TUBG1[A429-A432-A435]-CCC cells was reduced compared with *TUBG1*-sgRNA-U2OS-TUBG1-CCC cells (Supplementary Fig. 10).

To investigate whether the decreased colony-forming ability noted for the *TUBG1*-sgRNA-U2OS-TUBG1[A429-A432-A435] cells was caused by a slower progression of the S phase, we

presynchronized *TUBG1*-sgRNA-U2OS-TUBG1 and *TUBG1*-sgRNA-U2OS-TUBG1[A429-A432-A435] cells with thymidine and examined the cell cycle profiles of the synchronous populations[4]. Compared with the cells expressing γ-tubulin[wildtype], the *TUBG1*-sgRNA-U2OS-TUBG1[A429-A432-A435] cells exhibited delayed S phase progression (Fig. 8a), which demonstrates that mutation of the γ-tubulin PIP motif restricts execution of the S phase.

Finally, in comparison with γ-tubulin[wildtype] or with cells treated with colcemid, the expression of γ-tubulin[A429-A432-A435]

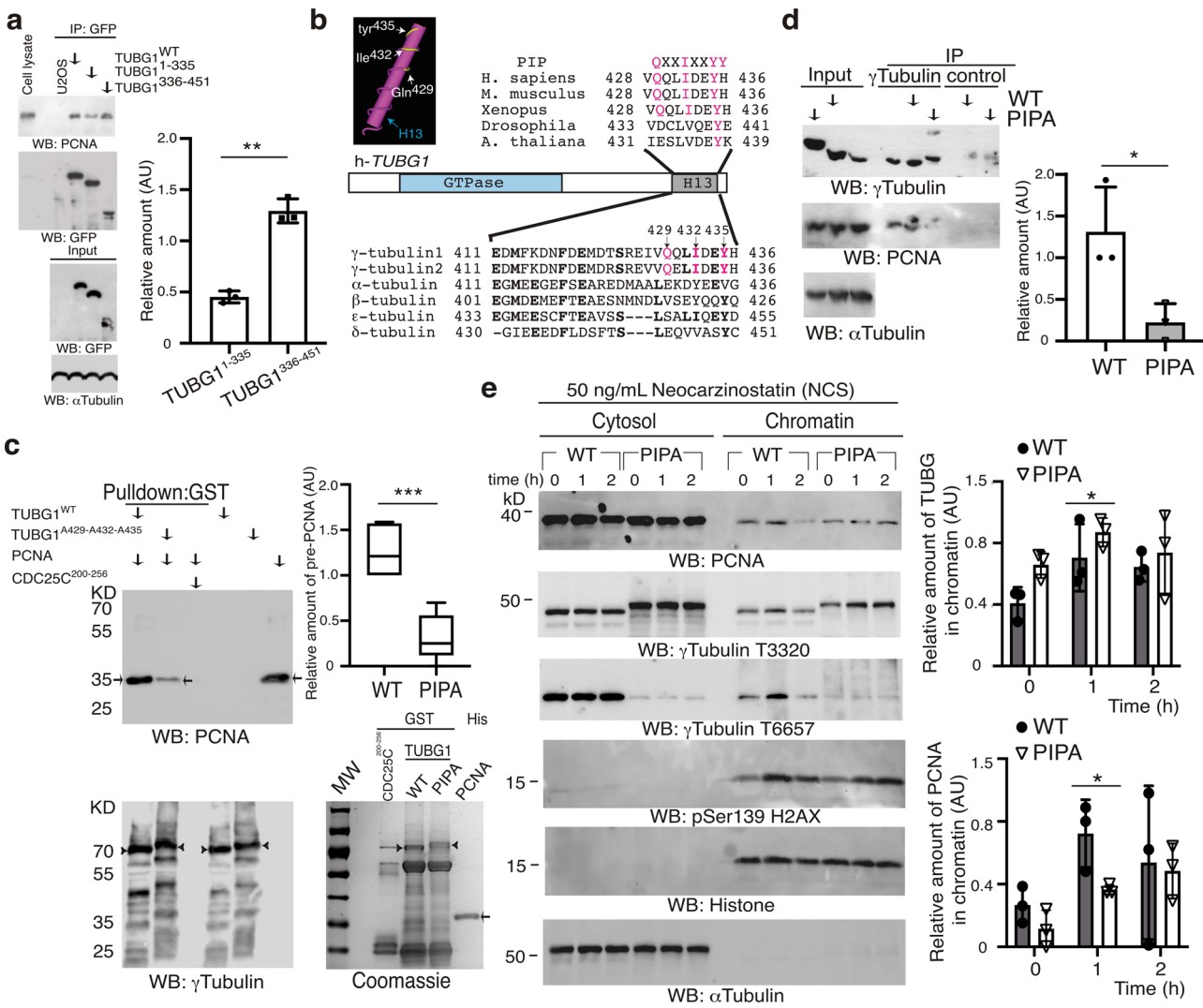

**Fig. 6 The γ-tubulin C terminus interacts with PCNA. a** Extracts from U2OS cells, or such cells stably coexpressing *TUBG*-shRNA and GFP-γ-tubulin, GFP-N-γ-tubulin[1–335], or GFP-C-γ-tubulin[336–451] were used, and GFP was immunoprecipitated with an anti-GFP antibody and developed by WB with the indicated antibodies ($N = 3$). **b** Sequence alignment of the C-terminal α-helix (H13) region of human γ-tubulin 1 and γ-tubulin 2 and α-, β-, ε-, and δ-tubulin. Bold letters indicate identical residues. Magenta letters represent residues included in the PIP. The known 3D structure of the C-terminal H13 region of human γ-tubulin revealed with the 3D structure viewer Cn3D. Important amino acids are highlighted in white. **c** Purified His-PCNA (arrows) was incubated with one of the following glutathione Sepharose affinity resin-bound GST-tagged proteins (arrowheads): γ-tubulin (TUBG1[WT]; WT), γ-tubulin[A429-A432-A435] (TUBG1[A429-A432-A435]; PIPA), or CDC25C[200–256]. Thereafter, the bound His-PCNA was examined by WB and the mean value of the PCNA signal (determined as its ratio with WT or PIPA) is represented in the graph (mean ± SD; $N = 5$, ***$P < 0.001$). Total amounts of loaded proteins were analyzed by WB and Coomassie staining. **d** Extracts from U2OS cells or U2OS cells co-expressing stably *TUBG1*-sgRNA and either γ-tubulin_resist (WT) or a γ-tubulin[A429-A432-A435]_resist (PIPA) were used, and γ-tubulin was immunoprecipitated with an anti-γ-tubulin antibody and developed by WB with the indicated antibodies ($N = 3$). **a, d** One-tenth of the total lysates used for immunoprecipitation was analyzed by WB (Input). The graphs illustrate densitometric analysis of the PCNA content in the indicated protein immunoprecipitates in the presented WBs. To adjust for differences in protein loading, the concentration of a protein was determined as its ratio with the immunoprecipitated protein ($N = 3$). **e** Extracts from neocarzinostatin-treated U2OSsg*TUBG1*-TUBG1[WT] (WT) and U2OSsg*TUBG1*-TUBG1[A429-A432-A435] (PIPA) cells were prepared as in Fig. 2b and examined by WB using the indicated antibodies against PCNA, γ-tubulin, phospho-histone2AX, α-tubulin, and histone. The two graphs illustrate the mean values of the γ-tubulin and the PCNA signal in chromatin. To adjust for differences in protein loading, the concentration of a protein was determined as its ratio with histone for each time point (mean ± SD; $N = 3$, *$P < 0.05$). **a, c–e** Source data are provided in Supplementary Data 1. See also Supplementary Fig. 8.

did not alter interphase microtubules or astral microtubule regrowth in *TUBG1*-sgRNA-U2OS-TUBG1[A429-A432-A435] cells, suggesting that the effects of γ-tubulin[A429-A432-A435] on cell cycle progression and PCNA recruitment to chromatin are not due to defective microtubule dynamics (Supplementary Fig. 11).

**The PCNA-loading replication factor C binds to chromatin in *TUBG1*-sgRNA-U2OS-TUBG[A429-A432-A435] cells.** To determine whether expression of the γ-tubulin PIP motif acts upstream of

the chromatin loading of RFC3 (loads PCNA onto chromatin)[34,46], we examined the formation of the pre-replication complex and the recruitment of RFC3 in synchronized *TUBG1*-sgRNA-U2OS-TUBG1 and *TUBG1*-sgRNA-U2OS-TUBG1[A429-A432-A435] cells. In contrast to *TUBG1*-sgRNA-U2OS-TUBG1 cells, *TUBG1*-sgRNA-U2OS-TUBG1[A429-A432-A435] cells had decreased levels of cyclin A (G2 marker) and cyclin D1 (G1 marker, indicates the start of the next G1), whereas the protein levels of Mcm5 and Mcm6 (license DNA replication) were

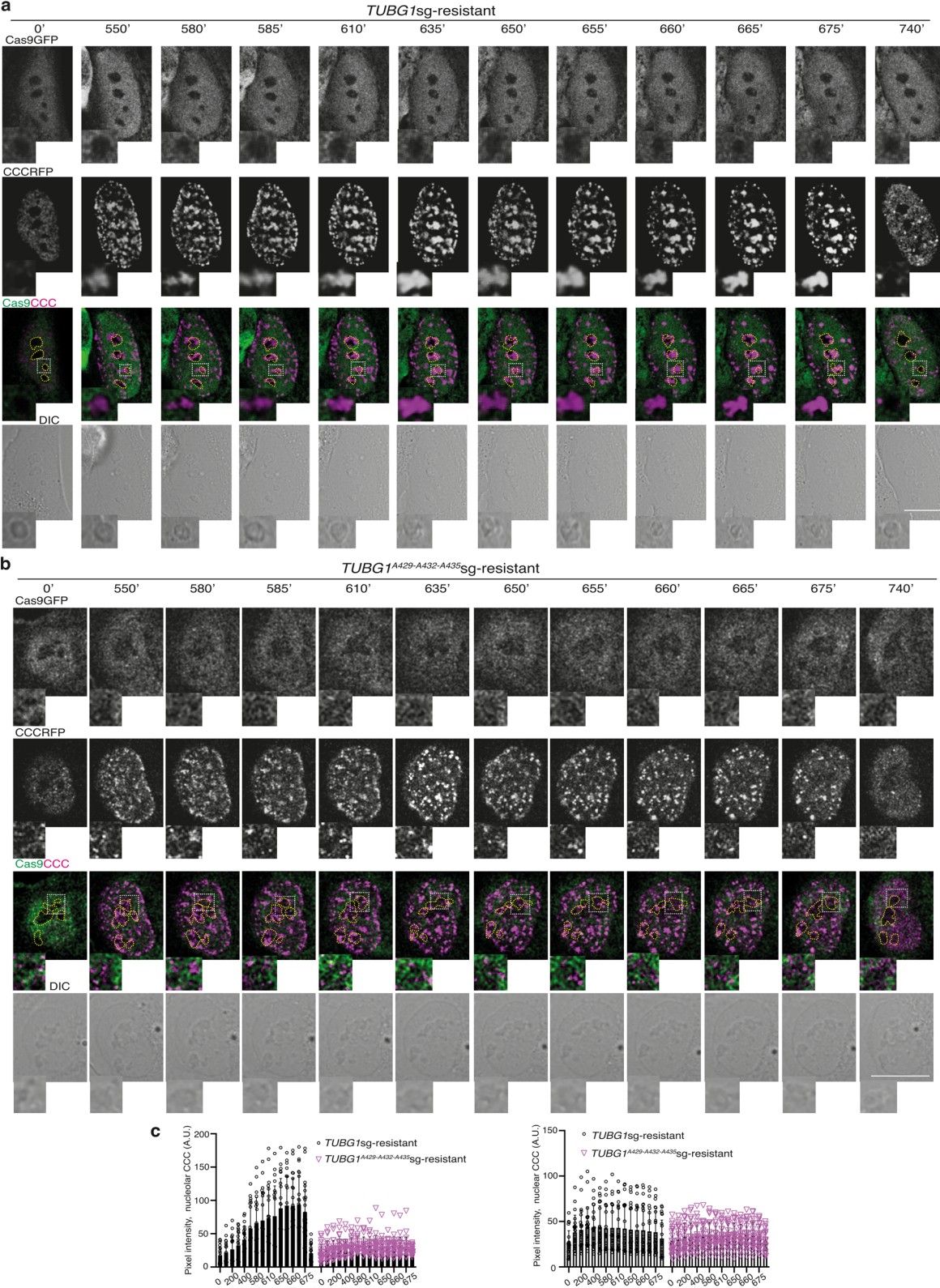

**Fig. 7 The γ-tubulin PIP mutant reduced the size of CCC complexes formed in the nucleus. a, b** The DIC/fluorescence images show time-lapse series of a stable *TUBG*sg-U2OS cell co-expressing either γ-tubulin$_{resist}$ (**a**) or a γ-tubulin$^{A429-A432-A435}_{resist}$ (**b**) and cell cycle chromobody (CCCRFP). The image series present chosen frames illustrating the accumulation of CCCRFP and how mutations in the γ-tubulin PIP motif affect the formation of CCCRFP foci (*N* = 30 cells). Scale bars: 10 μm (see also Supplementary Fig. 9). **c** The graphs show the time-dependent changes in fluorescence intensity in the nucleolus and nuclear compartment as indicated by the cells in **a**, **b** (AU; mean ± SD; *N* = 22 cells, ***P* < 0.001). Source data are provided in Supplementary Data 1.

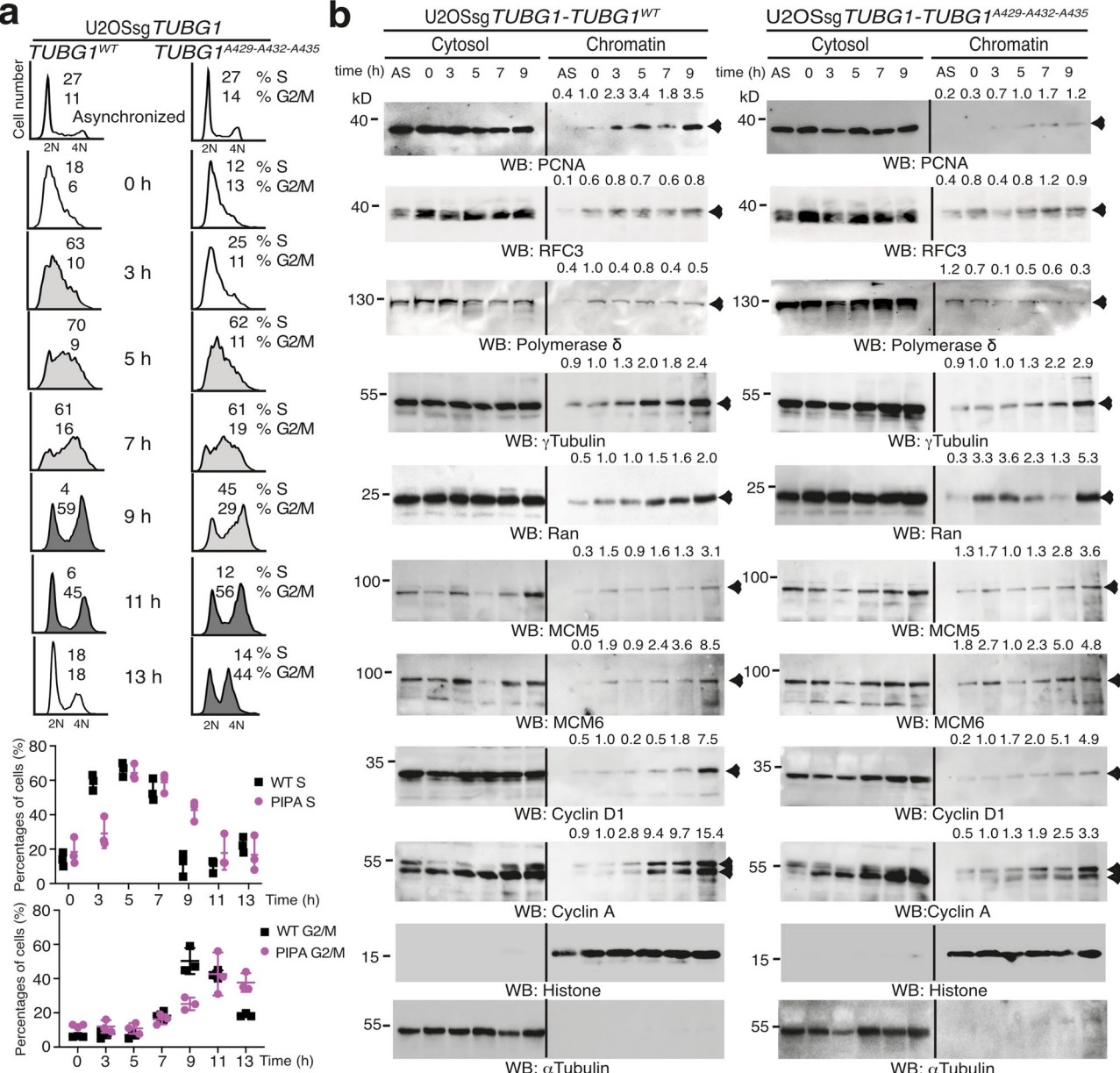

**Fig. 8 The γ-tubulin PIP mutant does not prevent the licensing of DNA replication. a, b** Analysis of the DNA (**a**) and protein (**b**) content in asynchronized (AS) and double thymidine block-synchronized *TUBG*sg-U2OS (U2OSsg*TUBG1*) cell line co-expressing single-guided (sg)-resistant γ-tubulin (TUBG1[WT]) or a γ-tubulin[A429-A432-A435] (TUBG1[A429-A432-A435]). **a** The data on each cell population are presented as the proportion of cells in S phase (<40% [black open histograms], >40% [light gray closed histograms]), or G2/M phase (>40% [dark gray closed histograms]). The graphs show the percentage of cells in S or G2/M phase (mean ± SD; $N = 3$, ****$P < 0.0001$; see also Supplementary Figs. 2, 10, and 11). **b** U2OSsg*TUBG1*-TUBG1[WT] and U2OSsg*TUBG1*-TUBG1[A429-A432-A435] cells were lysed, and proteins were biochemically divided as in Fig. 2b and then analyzed by WB with the indicated antibodies. α-Tubulin and histone were used as molecular markers for the cytosolic and nuclear fractions, respectively. Arrowheads indicate protein for which accumulation in the chromatin fraction varies during cell division. Numbers on the WBs indicate relative variations in accumulation of the proteins PCNA, RFC3, DNA polymerase δ, γ-tubulin, Ran, MCM5, MCM6, cyclin D, and cyclin A in the chromatin fraction. To adjust for differences in protein loading, for each sample, the concentration of an individual protein was determined by its ratio with histone ($N = 3$). **a, b** Source data are provided in Supplementary Data 1.

increased in the chromatin fraction (Fig. 8b)[45], suggesting that the pre-replication complex that licenses replication origins during early G1 phase is unaffected in *TUBG1*-sgRNA-U2OS-TUBG1[A429-A432-A435] cells[45]. Also, the levels of PCNA in the chromatin were decreased in *TUBG1*-sgRNA-U2OS-TUBG1[A429-A432-A435] cells despite an enrichment of γ-tubulin[A429-A432-A435], RFC3, and the presence of DNA polymerase δ in the chromatin fraction (5, 7, and 9 h; Fig. 8b). Finally, the chromatin

accumulation of Ran (which regulates nucleocytoplasmic transport) occurred earlier (i.e., at 0 and 3 h) in *TUBG1*-sgRNA-U2OS-TUBG1[A429-A432-A435] cells (Fig. 8b), which demonstrates a change in the balance between the cytoplasm and the nucleoplasm. Together, these data suggest that the pre-replication complex, RFC3, and DNA polymerase δ are loaded in sgRNA-U2OS-γ-tubulin[A429-A432-A435] cells, further confirming that the slower progression through S phase of *TUBG1*-sgRNA-U2OS-

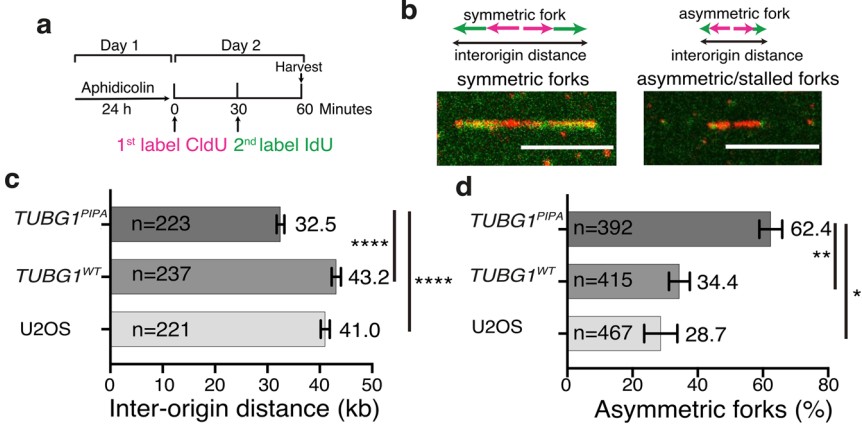

**Fig. 9 A decreased replication fork progression in U2OSsg*TUBG1*-*TUBG1*^A429-A432-A435 cells. a** U2OS, *TUBG1*-sgRNA-U2OS-TUBG1 (TUBG1^WT), and *TUBG1*-sgRNA-U2OS-TUBG1^A429-A432-A435 (TUBG1^PIPA) cells were treated with aphidicolin for 24 h, followed by pulse labeling with chloro deoxyuridine (CldU) for 30 min and sequentially with iodo-deoxyuridine (IdU) for 30 min. **b–d** The confocal images show DNA fibers containing stained symmetric and asymmetric forks and interorigin distance. Scale bars: 10 μm. The graphs show the interorigin distance (initiation rate) and the percentage of asymmetric forks (stalled forks) in the indicated cell line (mean ± SD; ****$P < 0.0001$, **$P < 0.01$).

TUBG1^A429-A432-A435 cells is caused by the reduced accumulation of PCNA in chromatin.

**Slow fork progression and shorter interorigin distance in *TUBG1*-sgRNA-U2OS-TUBG^A429-A432-A435 cells.** To examine the consequences of an impaired recruitment of PCNA on DNA elongation and on firing of dormant origins, we induced replication fork stalling by treating cells with aphidicolin, a DNA polymerase inhibitor that causes the firing of dormant origins of replication (Fig. 9a), and analyzed fork progression and dormant domain activation with a DNA-fiber assay (Fig. 9). This method entails the cellular labeling of nascent DNA, followed by the spreading of DNA fibers on microscope slides in U2OS, *TUBG1*-sgRNA-U2OS-TUBG1, and *TUBG1*-sgRNA-U2OS-TUBG1^A429-A432-A435 cells[52] (Fig. 9b). Analysis of the length of stained tracks showed a decreased replication fork progression in *TUBG1*-sgRNA-U2OS-TUBG1^A429-A432-A435 cells compared to U2OS or *TUBG1*-sgRNA-U2OS-TUBG1 cells, tracks of which were 21 and 25% longer, respectively (Fig. 9c). We found that the average interorigin distance in U2OS and *TUBG1*-sgRNA-U2OS-TUBG1 cells was 41.0 and 43.2 kb, respectively, which was significantly larger than the 32.5 kb observed in *TUBG1*-sgRNA-U2OS-TUBG1^A429-A432-A435 cells (Fig. 9c). Both the decreased interorigin distance (Fig. 9c), indicative of the initiation rate, and the increased number of asymmetric forks (Fig. 9d), a hallmark of stalled forks, found in cells expressing the PIP mutant reflect the reduced ability of the mutant to load PCNA onto origins of replication. A reduced loading of PCNA to fired dormant origins slow fork progression, and cells fire additional dormant replication origins to rescue replication, collectively causing a reduced interorigin distance. Altogether, the data show that the interaction between PCNA and γ-tubulin is required for replication fork progression.

**High expression of γ-tubulin is correlated with high expression of PCNA in tumors.** To test a possible synergism of PCNA and γ-tubulin in the maintenance of indefinite proliferation and development of cancer, we evaluated a potential interrelationship between *PCNA* and *TUBG1* in different tumor types. We used a large number of publicly available datasets from The Cancer Genome Atlas (http://cancergenome.nih.gov) and analyzed them with the GEPIA software[53]. Consistent with our hypothesis, we found a significant positive correlation between *TUBG1* and

*PCNA* expression in all the datasets we examined (33 tumor subtypes, $N = 9,664$; Supplementary Fig. 12, and Supplementary Table 2). The Pearson correlation coefficient ($R$) for 15 of the tumor subtypes we studied was ≥0.5, demonstrating a strong correlation between the expression of *TUBG1* and *PCNA*. In contrast, no significant correlation between the expression levels of *TUBG2* and *PCNA* was found for 17 of the investigated tumors subtypes, whereas 7 and 9 tumor subtypes, respectively, showed a week negative and a positive correlation between *TUBG2* and *PCNA* expression ($R ≤ 0.5$; Supplementary Fig. 12 and Supplementary Table 2).

To find out whether the mRNA expression represents TUBG1 and PCNA regulation at the protein level, we studied a possible interdependency between the levels of γ-tubulin protein and PCNA and the impact of such interdependence on the overall survival of patients in an ovarian cancer cohort (Fig. 10a, b)[54]. We divided the cohort into two subgroups according to the levels of γ-tubulin protein: a low-TUBG group (57 patients) and a high-TUBG group (90 patients) (Fig. 10a). In the group with elevated expression of γ-tubulin, we found that ~88% of the patients (79/90) also had an increased PCNA expression; in contrast, in the group of patients with a low γ-tubulin staining, ~67% of the tumors (38/57) showed a strong PCNA staining (38/57; Fig. 10b). A chi-squared test confirmed that there was a significant association between the expression of γ-tubulin and PCNA staining in the studied cohort ($P$ value of the chi-squared test $2.2 × 10^{-3}$) (Fig. 10b). In addition, we found that neither γ-tubulin nor PCNA occurred at levels that were associated with the overall survival of patients (log-rank $P$ values 0.82549 and 0.2233, respectively).

Together, the results outlined above demonstrate that there is a positive correlation between the expression of γ-tubulin and PCNA in tumors, which agrees with both the gene expression data that we analyzed and the knowledge that both these proteins are involved in cell proliferation.

**Discussion**
Here we show that the nuclear accumulation of γ-tubulin during S phase progression and in response to DSBs facilitates the transport of cytosolic PCNA to the nucleus and to active origin of replication. PCNA is associated with γ-strings in centrosomes[55] and γSNBs[22] and with γ-strings that span from the cytosolic compartment through the membranes and into the chromatin[16].

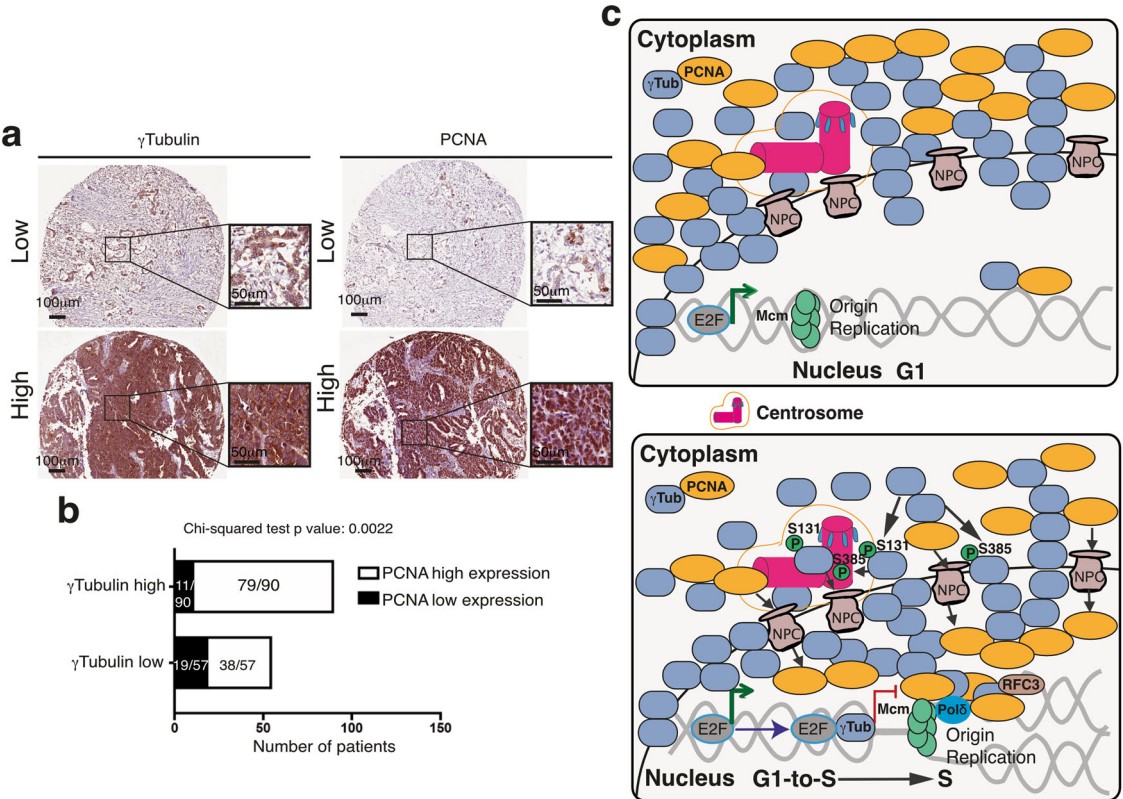

**Fig. 10 Expression of γ-tubulin and PCNA in an ovarian cancer cohort. a** Representative immunohistochemical staining of γ-tubulin and PCNA proteins. Staining is denoted as low or high. **b** Graph showing the relationship between the percentage of nuclei stained with PCNA and the levels of expression of γ-tubulin (see also Supplementary Fig. 12 and Supplementary Table 2). **c** Schematic representation of the different ways in which the γ-tubulin meshwork can control DNA replication. During nuclear formation, γ-tubulin establishes a nuclear protein boundary around chromatin that connects the cytoplasm and the nuclear compartment together. In G1 phase, the minichromosome maintenance complexes (MCMs) together with additional proteins are recruited to the origin of replication forming the pre-replication complex[45]. In S phase, the phosphorylation of γ-tubulin simultaneously initiates centrosomal duplication and increased nuclear γ-tubulin levels[2-4,20-24]. The E2F–γ-tubulin complex forms and terminates the transcriptional activity of E2Fs (red and green lines indicate inhibition and activation, respectively)[2]. The transcriptionally repressed chromatin is required to activate the licensed origins of replication (loaded with MCMs) and for the recruitment of PCNA to replication factor C complex (RFC3)-loaded chromatin by the γ-tubulin meshwork and the nuclear pore complexes (NPC)[34,46]. In the chromatin, PCNA encloses DNA and increases the efficiency of replicative polymerases, such as DNA polymerase delta (pol δ).

We found that reduced levels of γ-tubulin result in accumulation of PCNA in the cytoplasm, whereas expression of a PIP mutant affects the formation and the size of PCNA–γ-tubulin complexes and consequently attenuates the accumulation of PCNA in chromatin. Despite a decreased transport of PCNA, we observed accumulation of the DNA replication licensing factors Mcm5 and Mcm6, the PCNA chromatin-loading complex RFC, and DNA polymerase δ in the chromatin of cells expressing the PIP mutant, but the replication of individual DNA fibers was shorter. This implies that the reduced interaction of the PIP mutant with PCNA impairs the accumulation of PCNA in chromatin and slow down the replication rate during S phase execution, providing a novel mechanistic explanation for how PCNA present in the cytosol is recruited to chromatin.

Only one of the four antibodies tested recognized the unique conformation of chromatin-associated γ-tubulin[4]. Our analysis also revealed that the size of the nuclear pool of γ-tubulin is decreased in differentiated NSCs, implying that chromatin-associated γ-tubulin is necessary for maintenance of the proliferative potential. Accordingly, we found a positive correlation between the RNA expression of *TUBG1* and *PCNA* in all types of tumors we investigated and also between the expression of both proteins in an ovarian cancer cohort, denoting a strong interrelationship between the γ-tubulin meshwork and PCNA in various tumors.

Removal or disruption of the centrosomes or decreased levels of γ-tubulin cause G1 arrest[2,5,11,48,56]. In addition, we find that the PCM shuttles between chromatin-associated CCC foci. Considering these observations, along with the many reports describing that the PCM acts as a signal hub that brings together various checkpoint proteins during cell division and DNA damage response[57], it seems that γ-strings in centrosomes[55], nuclear envelope[16], nucleoli[22], and chromatin[2,3,20,22] may serve: first, as structuring elements that assist in the transport of cytosolic proteins to certain nuclear domains such as origin of replication or nucleoli, and second, as docking and sculpturing tools for the expansion of protein foci during DNA replication. In support of this hypothesis, reduced levels of γ-tubulin or mutations in the γ-tubulin PIP motif lead to accumulation of PCNA in the cytoplasm and a subsequent reduction in the amount of chromatin-associated PCNA.

Regarding the mechanism by which γ-tubulin accumulates in chromatin, we believe that it may involve the constant changes in the positioning of the centrosomes, together with the phosphorylation of γ-tubulin[4,23]. The later event reduces astral microtubule nucleation at the centrosomes and initiates a conformational change that unmasks the NLS, prompting the transport of γ-tubulin to the nucleus. This chain of events, together with the constant changes in the positioning of the centrosomes on the

surface of the nuclear envelope[5], will aid the delivery of γ-tubulin to necessary chromatin domains.

Some of the signals that synchronize centrosome duplication with DNA replication arise from signal transduction events that occur at the centrosomes. One such event is the translocation of Orc from the origin of replication to the growing centrosome[58]. In vertebrates, the sites of initiation of replication are not random and consist of binding sites for various transcription factors and for the protein PURA[44]. Sequencing of the DNA associated with PCNA and γ-tubulin immunoprecipitates from S phase-synchronized cells showed that, in the γ-tubulin peaks that overlapped with PCNA peaks, the most significantly enriched motif is the binding site for PURA[42,43]. This finding implies that, by binding to the PURA motif, γ-tubulin may control both DNA replication and transcription. This notion was confirmed by the overlapping found in the DNA associated with PCNA, Mcm5, FoxM1, and γ-tubulin immunoprecipitates and by earlier findings showing that, in S phase, γ-tubulin turns off the activities of the transcription factor E2F1[2]. Together, these observations indicate that, in early S phase, the accumulation of γ-tubulin in active origins of replication and the initiation of centrosomal duplication lead to the following events, all of which give PCNA access to the replication machinery: recruitment of Orc by the duplicating centrosome to activate origins of replication, turning off of the transcriptional activities of E2F1, and recruitment of cytosolic PCNA to the nuclear compartment (Fig. 10c).

Our results demonstrate a novel function of the γ-tubulin meshwork as a dynamic scaffold that facilitates the transport of PCNA into the nuclear compartment. Our findings provide a logical explanation for the mechanism that coordinates the turning off of transcription upon initiation of both DNA replication and centrosomal duplication[2,4,23]. In addition, our observations reveal a mechanism for targeting the nuclear activity of PCNA that involves interfering with the transport-assisting function of the γ-tubulin meshwork in tumor cells.

## Methods

**Reagents and cDNA**. The following antibodies and reagents were used: CCC encoding for anti-PCNA V$_H$H (anti-PCNA nanobody) fused to RFP tag (Chromotek, Germany); human C-terminal mCherry tagged TUBG2/pReceiver-M56 (GeneCopoeia, Rockville, USA); anti-GFP (1:500, mouse sc-53882 and rabbit sc-8334 antibodies), Ran (1:500), Mcm5 (1:500), Mcm6 (1:500), RFC3 (1:500), lamin B1 (1:1500), TCP1β (1:500), γ-tubulin (1:1000, mouse, sc17788), β3-tubulin (1:500), Sox2 (1:500), nucleolin (1:2000), anti-PCNA (1:500, horseradish peroxidase-tagged mouse sc-56 and rabbit sc-7907; used for western blotting), DNA polymerase δ (1:500), and GCP2 (1:500, from Santa Cruz Biotechnology, Dallas, TX, USA); anti-βactin (1:5000), anti-pericentrin (1:500) and anti-γ-tubulin (1:500, mouse T6557 [used for western blotting and immunoprecipitation], and rabbit T3320 [used for western blotting and immunofluorescence] and T5192 [used for western blotting] antibodies); Sigma-Aldrich, St. Louis, MO, USA); anti-PCNA (1:1000, mouse; used for western blotting, immunofluorescence, and immunoprecipitation), anti-pSer[139] H2AX (1:1000; a marker for DNA damage) and anti-α-tubulin (1:1000, mouse, Millipore, CA, USA); anti-histone (1:400, Merck); anti-α-tubulin (1:1000, rabbit, Cell Signaling Technology). All other reagents were obtained from Sigma-Aldrich.

The anti-γ-tubulin antibodies recognized the following γ-tubulin sequences: T3320, raised against amino acids mapping at the C-terminus of Xenopus γ-tubulin; T6557, raised against amino acids 38–53 mapping at the N-terminus of γ-tubulin; sc17788, raised against amino acids 269–451 mapping at the C-terminus of human γ-tubulin; and T5192, raised against amino acids 38–53 mapping at the N-terminus of γ-tubulin.

Human TUBG shRNA, pEGFP-sh-resistant TUBG1, TUBG1[1–335], and TUBG1[336–451] genes, pSpCas9(BB)-2A-GFP TUBG1 sgRNA, pGEX2T-TUBG1, and pcDNA3-sg-resistant TUBG1 were prepared as previously described[2,23,59]. Removal of the internal TUBG2-EcoRI site in pReceiver-M56-TUBG2-mCherry and the Gln429-to-Ala–Ile432-to-Ala–Tyr435-to-Ala substitutions in pcDNA3-sg-resistant TUBG1 and pGEX2T-TUBG1 genes were prepared using a Quickchange Mutagenesis Kit (Stratagene) and the primers listed in Supplementary Table 3. The sgRNA for human PCNA was cloned into the pSpCas9(BB)-2A-GFP (PX458) plasmid (a gift from Dr. F. Zhang, MIT[60]; Addgene plasmid 48138) with the annealed oligonucleotides listed in Supplementary Table 3. The TUBG2 gene lacking its internal EcoRI site was amplified by PCR and subcloned in frame into

EcoRI/EcoRV of pcDNA3.1 (Thermo Fisher Scientific, MA, USA) using the oligos listed in Supplementary Table 3. The mutations and constructs were verified by sequencing.

**Cell culture, transfection, and western blot analysis**. H9-NSCs were a gift from Dr. L. Karayan-Tapon (Poitiers University Hospital, France[61]) and were cultured at 37 °C and 5% CO$_2$ in knock-out Dulbecco's Modified Eagle Medium (DMEM)/F-12 medium supplemented with 2% Stempro neural supplement, 20 ng/mL recombinant basic fibroblast growth factor, 20 ng/mL recombinant epidermal growth factor (Gibco™), 2 mM Glutamax, 100 U/mL penicillin, and 100 mg/mL streptomycin; this was done on Geltrex-coated culture dishes (Gibco™; coating of plates was performed according to the manufacturer's instructions). Cells were passed using the StemPro Accutase solution (Gibco™). Neuronally differentiated H9-NSCs were obtained by culturing H9-NSCs for 14 days in neural induction media (DMEM Neurobasal media, Gibco™) supplemented with serum-free B27 supplement, 2 mM Glutamax, 100 U/mL penicillin, and 100 mg/mL streptomycin. To assay neuronal differentiation, changes in the expression of Sox2 (NCS marker) and β3-tubulin (neuronal marker) and in cell morphology were analyzed by western blotting and microscopy. H9-NSCs express γ-tubulin 1 and γ-tubulin 2. NIH3T3, U2OS, MCF10A, and A549 cells were cultured and transfected, and cell lysates and immunoprecipitations were prepared and analyzed by western blotting as previously described[16,20,23,62,63]. NIH3T3, U2OS, MCF10A, and A549 express γ-tubulin 1.

U2OS cells were transfected with sgRNA and examined on day 10[11]. Stably or transient transfected TUBG1-sgRNA-U2OS, TUBG1-sgRNA-U2OS-TUBG1, TUBG1-sgRNA-U2OS-TUBG2, PCNA-sgRNA-U2OS, TUBG-shRNA-U2OS, TUBG-shRNA-U2OS-TUBG1-GFP, TUBG-shRNA-U2OS-TUBG1[1–335]-GFP, TUBG-shRNA-U2OS-TUBG1[336–451]-GFP, TUBG-shRNA-U2OS-TUBG1-CCC, TUBG-sh-MCF10A, TUBG1-sgRNA-U2OS-TUBG1[A429-A432-A435], TUBG1-sgRNA-U2OS-TUBG1-CCC, and TUBG1-sgRNA-U2OS-TUBG1[A429-A432-A435]-CCC cells were obtained as described in Supplementary Table 4 and elsewhere[62]. Stably transfected cells were obtained by selection with 100 μg/mL zeocin (Invitrogen) or 200 μg/mL G418, or by combining zeocin and G418, as previously described[62].

**Fractionation and immunoprecipitation**. Biochemical fractions and total lysates of cells were prepared as reported elsewhere[2,4]. Briefly, cells ($1.0 \times 10^6$) were lysed in BAD buffer (10 mM HEPES [pH 7.9], 10 mM KCl, 10% glycerol [v/v], 1.5 mM MgCl$_2$, 340 mM sucrose, 1 mM dithiothreitol (DTT), 250 mM phenylmethylsulfonyl fluoride [PMSF], and 100 mM Na$_3$VO$_4$, 0.5 mg/mL aprotinin, 0.5 mg/mL leupeptin, and 0.5 mg/mL pepstatin) containing 0.1% Triton X-100[64], and the supernatant constituted the cytosolic fraction. The pellet containing nuclei was lysed in NSB buffer (3 mM EDTA and 0.2 mM EGTA dissolved in water[64]). Thereafter, the supernatant of lysed nuclei was the nuclear membrane fraction, and the remaining pellet was the chromatin fraction. DNA in the chromatin fraction was degraded by treatment with micrococcal nuclease, as stipulated by the manufacturer (New England Biolabs). The purified fractions from $1.0 \times 10^6$ cells were boiled in sample buffer and analyzed by western blotting using α-tubulin and histone as molecular markers for the cytosolic and nuclear fractions, respectively. Unless otherwise indicated, as microtubules and other cytosolic content bind to the nuclear membrane, we mixed the cytosolic and the nuclear membrane fraction to create a single fraction that was subsequently called the cytosolic fraction. Total lysates were prepared by pooling together the cytosolic, the nuclear membrane, and the chromatin fraction. In this way, all cellular fractions were analyzed.

Biochemical fractions of U2OS cells ($20 \times 10^6$) in the absence of detergent were prepared as follows. Initially, to remove possible cytoskeletal elements attached to the nuclear envelope, cells were pre-incubated for 20 min in culture medium containing 100 ng/mL colcemid and 5 μg/mL cytochalasin B (37 °C, 5% CO$_2$)[5,16,39]. Once harvested, the cells were lysed on ice in a dounce homogenizer in 0.7 mL cold BAD buffer. The cell debris, unbroken cells, and nuclei were removed by centrifugation ($1300 \times g$ for 5 min at 4 °C). The nuclei were resuspended in 500 μL of BAD buffer and layered onto a 250-μL of a 20% sucrose cushion in BAD buffer and centrifuged ($3400 \times g$ for 5 min at 4 °C) to remove cell debris[16]. The nuclear membrane and chromatin fractions were obtained as described above. DNA in cytosolic and chromatin fractions was degraded by treatment with micrococcal nuclease. The final fractions were adjusted to a volume of 1 mL and final concentrations of 340 mM sucrose and 0.5% Triton X-100 before layering each sample onto a discontinuous gradient consisting of 500, 300, and 300 μL of 70%, 50%, and 40% sucrose, respectively, dissolved in a solution containing 10 mM HEPES [pH 7.9], 0.5% Triton X-100, 1 mM DTT, and 250 mM PMSF before centrifugation ($120,000 \times g$ for 45 min at 4 °C in a swing out rotor, S52-ST).

The chromatin fractions were treated with micrococcal nuclease. Then the final cytosolic and chromatin fractions were sonicated three times for 5 s before being subjected to immunoprecipitation. Briefly, the final fractions were adjusted to a volume of 500 μL and 1% Triton X-100, Tris 20 mM (pH 7.5), 150 mM NaCl, 1 mM EGTA, 1 mM EDTA, 1 mM β-glycerolphosphate, 2.5 mM sodium pyrophosphate, 0.1 mM Na$_3$VO$_4$, 1 μg/mL leupeptin, and 1 mM PMSF final concentration. The following mouse monoclonal antibodies were used for immunoprecipitation: T6557 (γ-tubulin, Sigma-Aldrich), sc-53882 (GFP, Santa Cruz Biotechnology), and CBL407 (PCNA, Millipore). The immunoprecipitates were analyzed by western blotting as previously described[23].

**Synchronization, DSB formation, and colony-forming assay**. U2OS cells were synchronized in $G_0/G_1$ by maintaining cell confluence for 48 h. To arrest cells in the early S phase, we presynchronized U2OS, MCF10A, or A549 cells by treatment with 2 mM thymidine as previously described[4]. Cell cycle progression was monitored by determining the DNA content of cells with a NC-3000 nuclear counter as specified by the manufacturer (Chemometec A/S, Denmark).

DSBs were induced by treating cells with different concentrations of NCS or cisplatin for various times, after which the cells were harvested and analyzed. To assay the survival after exposure to NCS, cells were harvested after 30 min of treatment and washed. The colony-forming unit was determined by platting 1000 cells in a 36-mm well and incubating until >50 colonies could be observed in the control plate (this required 8–10 days). The cells were subsequently fixed for 20 min in 500 μg/mL crystal violet in 1% formaldehyde and 10% methanol diluted in phosphate-buffered saline and then washed several times in deionized water and then air dried.

**ChIP and ChIP-seq analysis**. ChIP and ChIP-seq analysis of the data obtained from U2OS, MCF10A, and *TUBG*-sh-MCF10A cells are described elsewhere[2,11], with the following modification: coprecipitated chromatin from U2OS cells was sequenced using the 5500W Series Genetic analyzer according to the manufacturer's instructions (Thermo Fisher Scientific, MA, USA). Alternatively, ChIP samples were obtained from U2OS and *TUBG*-sh-U2OS cells as described elsewhere[2,11], with the following modifications: DNA was fragmented with 1000 U micrococcal nuclease (New England Biolabs) per $50 \times 10^6$ cells for 30 min at 37 °C to obtain an average fragment size of ~200 bp, and cell lysates were precleared and immunocomplexes collected with Dynabeads (Invitrogen). Libraries from coprecipitated chromatin were prepared from 1 to 10 ng of ChIP DNA using the Smarter ThruPLEX DNA-seq Kit (cat# R400676, Takara) with HT dual indexes (cat# R400660, Takara). The library preparation was performed according to the manufacturer's instructions (guide#112219). The quality of the libraries was evaluated using the Fragment Analyzer from Advanced Analytical (AATI) using the DNF-910 Kit. The adapter-ligated fragments were quantified by quantitative PCR using the Library quantification Kit for Illumina (KAPA Biosystems) on a CFX384Touch (Bio-Rad) prior to cluster generation and sequencing. Chromatin was sequenced using the NovaSeq S1 flowcell, PE50 bp and v1 sequencing chemistry, and an Illumina NovaSeq 6000 instrument (NovaSeq control software v 1.7.0/ RTA v3.4.4) according to the manufacturer's instructions. Demultiplexing and conversion to FASTQ format was performed using the bcl2fastq2 (v2.20.0.422) software, provided by Illumina. Additional statistics on sequencing quality were compiled with an in-house script from the FASTQ-files, RTA, and BCL2FASTQ2 output files. ChIPs from U2OS, MCF10A, and *TUBG*-sh-MCF10A cells were performed using rabbit polyclonal antibodies: a mixture (1:1) of anti-γ-tubulin T3320 and anti-γ-tubulin T5192 (Sigma-Aldrich), anti-PCNA sc-7907 (Santa Cruz Biotechnology), and anti-macro histone 2A1 #4827 (Cell Signaling). ChIPs from U2OS and *TUBG*-sh-U2OS cells were performed using a monoclonal anti-PCNA clone PC10 MAB424 (Sigma-Aldrich) and the following rabbit polyclonal antibodies: a mixture (1:1) of anti-γ-tubulin T3320 and anti-γ-tubulin T5192, anti-Mcm5 ab17967 (Abcam), and anti-FoxM1 GTX102170 (Gene Tex).

PeakSeq (Version 1.31) was used for peak calling of the data obtained from U2OS cells, with an false discovery rate cut-off of 0.01[65]. We used two different methods to investigate whether there was a propensity for PCNA peaks and γ-tubulin peaks to be close to each other. In the first approach, we calculated the distance distribution between a γ-tubulin peak and the PCNA peak closest to it, which was centered around 0, as expected. Thereafter, for each of the analyzed chromosomes, we randomized the PCNA peaks, ensuring that both the total number of PCNA peaks in any chromosome and the width of the PCNA peaks remained unchanged, but placing the peaks in random positions on the same chromosome. Next, we re-plotted the distance distribution between a γ-tubulin peak and its closest PCNA peak. If the true distance distributions were much narrower than the distance distributions after the PCNA peaks were randomized, it was regarded as a clear propensity for the PCNA peaks to occur in locations close to the γ-tubulin peaks.

In the second method, we performed a permutation-based statistical test to evaluate the propensity for PCNA binding to occur in the same regions in the genome as γ-tubulin binding. A PCNA peak was said to overlap a γ-tubulin peak if at least 25% of its span overlapped the span of the γ-tubulin peak. The position of the identified peaks was randomized for 10,000 times, keeping the width of the peaks unchanged.

Significantly enriched motifs were identified among overlapping γ-tubulin and PCNA peaks from MCF10A ChIP-seq data at an $E$ value cut-off of 0.05. This was achieved using the motif-based sequence analysis tool MEME-ChIP (version 5.1.0)[66].

Coprecipitated chromatin from U2OS, *TUBG*-shRNA-U2OS, and *TUBG*-shRNA-U2OS-TUBG1 cells was analyzed by PCR to assess the presence of the PURA DNA-binding motif between base pairs 49,109,512 and 491,097,020 in chromosome 4 with the oligos listed in Supplementary Table 3.

**Fluorescent imaging microscopy and microtubule regrowth assay**. Cells were cultured on coverslips and fixed as previously described[4,62,67]. The T3320 antibody

was used for immunofluorescence staining of γ-tubulin unless otherwise indicated. Confocal and fluorescence imaging were performed on a Zeiss LSM 700 Axio Observer microscope with a Plan-Apochromat ×40 or ×63 NA 1.40 oil immersion objective. All images included in this article that were captured with the mentioned microscope were subjected to a rolling ball background subtraction (Fiji). Sequential images were collected at 0.2- or 0.4-μm intervals. Time-lapse images were captured every 5 min. The time points shown in the figures represent minutes after initiation of the formation of nuclear CCC foci in the monitored cell. Cells were treated and analyzed as described previously[59]. Colocalization analysis, Z-stack projections, and processing of images were carried out with the ImageJ (Fiji) software. The plug-in "colocalization threshold" in Fiji was used to determine colocalization between two channels[68]. The Skeletonize three-dimensional (3D) plug-in (Fiji) was applied to generate the skeletonized stacks of cell images and to remove noise and binarize images[69]. The 3D viewer plugin (Fiji) was used to generate 3D surface images[70].

**Purification of recombinant proteins and reconstitution of the PCNA–γ-tubulin complex**. His$_6$-tagged PCNA was obtained from Sigma-Aldrich. The fragment of the cell division cycle 25C protein fused to GST (GST-CDC25C$^{200-256}$, a gift from Dr. H. Piwnica-Worms[71]) was expressed and purified as previously described[23]. The human GST fusion proteins (N-terminal-GST-fused γ-tubulin, GST-γtubulin$^{A429-A432-A435}$, and GST-CDC25C$^{200-256}$) were expressed in *Escherichia coli* DH5α. Exponentially growing bacteria bearing the remaining plasmids were first induced 1 h at 37 °C with 0.2 mM isopropyl-1-thio-β-D-galactopyranoside (IPTG) and then incubated in the presence of IPTG overnight at room temperature. Recombinant proteins were purified under native conditions using glutathione Sepharose 4B (Amersham Pharmacia Biotech) according to the manufacturer's instructions, with the following modifications: all buffers used were supplemented with 5 mM beta mercaptoethanol (βME), 1 mM MgCl$_2$, and 250 mM GTP.

GST-tagged proteins (500 ng) bound to 5–μL glutathione Sepharose 4B beads were incubated for 45 min with PCNA (500 ng) under rotation at 4 °C (total volume of 1 mL in buffer containing 20 mM Tris [pH 7.5], 0.1% Triton X-100, 1 mM MgCl$_2$, 5 mM βME, 137 mM NaCl, and 5% glycerol) and subsequently washed two times with 1 mL of buffer for 5 min under rotation. Thereafter, the beads were boiled and analyzed by western blotting.

**Fiber assay**. U2OS, *TUBG1*-sgRNA-U2OS-TUBG1, and *TUBG1*-sgRNA-U2OS-TUBG1$^{A429-A432-A435}$ cells were seeded at a density of $1.5 \times 10^5$ cells into 6-well plates. The cells were treated with 6 μM aphidicolin (Biochemica, #10797) for 24 h, followed by pulse-labeling with 25 μM chloro deoxyuridine (Sigma-Aldrich, C6891) for 30 min and sequentially with 250 μM iodo-deoxyuridine (Sigma-Aldrich, I7125) for 30 min. Collected cells were diluted to $5 \times 10^4$ cells/mL and then spread and stained as previously described[52]. Images were acquired with a Zeiss LSM 780 inverted confocal microscope and analyzed with the ImageJ software.

**Tissue microarray construction, immunohistochemistry, and analysis of co-expression of PCNA and γ-tubulin in various tumors**. We used the GEPIA software[53] to analyze the co-expression of *PCNA* and both *TUBG1* and *TUBG2* mRNAs in The Cancer Genome Atlas datasets.

Epithelial ovarian cancers tissues were obtained from 154 patients diagnosed at Malmö University Hospital, Malmö, Sweden. The cohort is a merge of all incident cases in the population-based prospective cohort studies entitled "Malmö Diet and Cancer Study," as described elsewhere[54]. For immunohistochemical analysis of γ-tubulin/PCNA, the tissue samples form the cohort were stained in an Autostainer Plus (DAKO, Glostrup, Denmark) with anti-γ-tubulin (T6557) and anti-PCNA (CBL407) antibodies diluted 1:2000. We divided the cohort into two subgroups according to the level of expression of γ-tubulin: a low-TUBG group (57 patients) and a high-TUBG group (90 patients). To investigate the association between γ-tubulin and PCNA, we studied the relationship between the percentage of nuclei positively stained for PCNA and the level of expression of γ-tubulin. A chi-squared test was used to determine the relationship between the expression of γ-tubulin and the expression of PCNA.

**Statistical and reproducibility**. All data are expressed as mean ± standard deviation, and the statistical significance of differences between two groups was analyzed by paired Student's $t$ test or by two-way analysis of variance using the Prism 8 software ($*P < 0.05$, $**P < 0.01$, $***P < 0.001$, $****P < 0.0001$). Cell cycle profiles were assessed using FlowJo (Tree Star, Inc.).

Different cultures from separate experiments were used to obtain data in each experiment. All source data underlying graphs in the main figures are available in Supplementary Data 1.

**Reporting summary**. Further information on research design is available in the Nature Research Reporting Summary linked to this article.

## Data availability

Raw sequence data and processed files are publicly available at NCBI Gene Expression Omnibus (GEO) (http://www.ncbi.nlm.nih.gov/geo) under the following accession

numbers: GSM2884568, GSM2884569, GSM2884570, GSM2884571, GSM2884576, GSM2884577, GSM2884578, GSM2884579, GSM2884584, GSM2884585, GSM2884586, GSM2884587, GSM2884588, GSM2884589, GSM2884590, GSM2884591, GSM2884572, GSM2884573, GSM2884574, GSM2884575, GSM2884580, GSM2884581, GSM2884582, GSM2884583, GSM5277562, GSM5277563,GSM5277564,GSM5277565, GSM5277566, GSM5277567, GSM5277568, GSM5277569, GSM5277570, GSM5277571, GSM5277572, GSM5277573, GSM5277574, GSM5277575, GSM5277576, GSM5277577, GSM5277578, GSM5277579, GSM5277580, GSM5277581, GSM5277582, GSM5277583, GSM5277584, GSM5277585, GSM5277586, GSM5277587, GSM5277588, GSM5277589, GSM5277590, GSM5277591, GSM5277592, GSM5277593, GSM5277594, GSM5277595, GSM5277596, GSM5277597, GSM5277598, GSM5277599, GSM5277600, GSM5277601, GSM5277602, GSM5277603, GSM5277604, GSM5277605, GSM5277606, GSM5277607, GSM5277608, GSM5277609, GSM5277610, GSM5277611, GSM5277612, GSM5277613, GSM5277614, GSM5277615, GSM5277616, GSM5277617, GSM5277618, GSM5277619, GSM5277620, GSM5277621, GSM5277622, GSM5277623, GSM5277624, and GSM5277625. All other data not presented in the manuscript or supporting materials are available from the corresponding author upon request. The source data underlying Figs. 1a–d, 2b–d, 3a–d, 4b, d, 6a, c–e, 7c, 8a, b, and 9c, d and Supplementary Figs. 1a, b, 2a–c, 3b, 4, 8a, b, and 10 are provided in Supplementary Data 1.

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

## Acknowledgements

We thank Dr. F. Zhang, Dr. H. Piwnica-Worms, and Dr. L. Karayan-Tapon for reagents; the SNP&SEQ Technology Platform in Uppsala that is part of National Genomics Infrastructure Sweden (NGI)/Science for Life Laboratory for help with massive parallel sequencing and computational infrastructure (funded by Swedish Research Council and the Knut and Alice Wallenberg Foundation); and Patricia Ödman for editorial assistance. This research was funded by the Swedish Cancer Society (grant number 190137 Pj), the Swedish Childhood Cancer Foundation (grant number PR2018-0083, PR-2019-0047, and PR-2020-0080), Skane University Hospital in Malmö Cancer Research Fund (grant number 20151209), the Royal Physiographic Society in Lund (grant number 2017-2019), and Lillian Sagen and Curt Ericsson's Research Foundation (to N.M.S.G.).

## Author contributions

M.A.K. conceived the project, designed the experiments, and performed some experiments. T.L. analyzed ChIP-seq data. M.C., J.Z., D.M., N.C., D.C., N.G., C.A.R., S.A., K.E.-H., N.C., K.J., and L.L. performed experiments and analyzed data. M.A.-K. wrote the paper with the inputs from all authors. All authors commented on and agreed with the content of this paper.

## Funding

## Competing interests

The authors declare no competing interests.
