## [Peer Review File · Communications Biology]

Reviewers' Comments:

Reviewer #1:

Remarks to the Author:

Corvaisier et al reports the interaction between gamma-tubulin and PCNA and its importance of chromatin loading of PCNA for proper cellular proliferation, especially in S phase and DNA damage response. Authors first isolated a proper antibody specifically detecting gamma-tubulin and used it for interaction study of PCNA. Authors rely on mainly cellular fractionation and microscopic observations to claim cytosolic PCNA is recruited to chromatin by gamma-tubulin. In addition, authors used chromatin immunoprecipitation and sequencing to show correlative accumulation of gamma-tubulin and PCNA in the genome. Lastly, authors analyzed pre-existing cancer patients data to support interdependent expression of gamma-tubulin and PCNA in various cancers. The manuscript contain somewhat surprising results and should be carefully examined whether all claims by authors are properly addressed in the manuscript. It will be important to address major as well as minor points listed below...

Major points.

It is little striking observation that PCNA was observed mainly in cytosolic fraction in U2OS cells authors used. Although there are several papers describing cytosolic PCNA (eg. J.Exp.Med (2010) 207:2631) which describes PCNA movement to the cytosol during differentiation, majority of publications so far describe high enrichment of PCNA in the nucleus where PCNA functions.

ChIP Seq analysis should have valid negative control which do not show correlation with gamma-tubulin such as transcription factor. In addition, to confirm these colocalizations indeed reflect DNA replication fork, positive control (DNA replication enzyme such as polymerase, RFC, etc) should be included.

Authors showed gamma-tubulin deficiency showed S phase problem (Fig. 2). It could be due to other defects caused by gamma-tubulin deficiency. How to exclude other defects caused by gamma-tubulin deficiency?

In Figure 4, authors claimed enrichment of gamma-tubulin at the origin of replication. It will be important to know whether such enrichment are always observed or changed in time dependent manner. It will be quite important to understand dynamics of gamma-tubulin for active, dormant, late firing origins.

Minor points.

Overall, figures are too small to look through them and words in the figures are hard to read (resolution problem as well as size).

In several places, "gamma-strings" are written as "gamma-Strings". They should be consistently written.

In Figure 1c, NSC and Neuron blot did not have any description what is the band means. Also, there appears to have several lanes, but not clear what each lane represents.

In Figure 1a, arrows and arrowheads are used. In the legend, arrowheads and arrows are described as gamma-tubulin 1 and 2. But, sgRNA additions are also indicated as arrows. Authors need to make figure marks clearly to distinguish them.

Figure 1C, the graph (top panel) labels are hard to follow.

Page 7, authors described "nocodazole" was used for experiment of Figure 1d. However, in the figure, "Colcemid" is described. Should be consistent between figure and text.

Figure 2a only shows one representing image. Quantification with more than 100 cells should be presented as graph.

Figure 3d, control 2 hour seems to have issue for loading. The representing blot should have all lanes properly showing necessary bands.

Page 11, "It should be noted that sg-mediated reduction of gamma-tubulin expression, also decreased the amount of gamma-tubulin associated with centrosomes, but ---" ; Does this mean gamma-tubulin in centrosomes have retention function of proteins in centrosome?

Reviewer #2:

Remarks to the Author:

The authors convincingly demonstrate that PCNA and g-Tubulin co-localize, and directly interact through a PIP domain in g-tubulin. These are important results which validate the need for functional investigations into the roles of this interaction. To this end, the authors perform immunofluorescence and chromatin fractionation experiments, concluding that g-tubulin facilitates PCNA chromatin recruitment. Even though it is a very important part of the manuscript, I found this aspect of the manuscript to be much less convincing. Critical experiments to address the impact of the interaction are not presented, instead the authors rely on correlative results. The immunofluorescence experiments lack quantifications from multiple replicates and are thus difficult to interpret. The experiments are not performed with multiple knockdown cell lines to rule out off-target effects.

Specific comments:

1. In Fig 2b, the quantification does not really match the blot. The upwards trend is clear on the blot, but not so much on the quantification.
2. A western blot showing the sgTub knockdown for the experiment in Fig3a is needed.
3. The authors used a single guide sequence and a single shRNA sequence throughout the manuscript. Some of the experiments are done with the guide, some are done with the shRNA, but no experiment is done with both of them to rule out off-target effects. To this end, they should use two independent sequences for each the guide and the shRNA.
4. The authors need to include some type of quantification, from multiple replicates, for the immunofluorescence experiments in Fig 3b, c and Fig 5. In the absence of this, it is very difficult to assess if the authors' conclusions are supported by the data.
5. Fig 3d should include the whole cell extracts as control.
6. Just from the chromatin fractionation shown in Fig 3d, it is difficult to conclude that PCNA chromatin localization depends on g-tubulin. This appears to be the main take home message of the manuscript, but it is not strongly supported by the data presented. The quantification shown in this figure shows a minimal difference, which is unlikely to be statistically significant. The authors should employ immunofluorescence experiments (with quantifications from multiple replicates) to investigate PCNA chromatin recruitment in shTUBG cells.
7. Page 11: it is confusing why the authors speculate that PCNA-tubulin complexes may limit the transport of PCNA to the nuclear compartment. Based on what do they speculate this? If anything,

they show that the complexes promote PCNA chromatin binding, which would be the opposite.

8. Fig 4a should include all the time points shown for gTub, also for PCNA and H2A. Moreover, multiple replicates of this experiment should be performed. Finally, to rule out off target effects, it is critical that the experiments in Fig 4 are performed with an independent shRNA knockdown sequence, or with the sg cell line. It is difficult to make any conclusions in the absence of this.

9. The CHIP-seq results (Fig4) do not show that "g-tubulin may facilitate the accumulation of PCNA in the nucleus", as it is wrongly stated on page 17. Instead, these results show that g-tubulin binds to the same DNA regions as PCNA is known to bind to. This would in fact imply the opposite, that PCNA recruits g-tubulin to those regions. It is important that the authors address this issue by investigating the ChIP-seq localization of PCNA to PURA sites upon g-tubulin depletion, and in the PIP mutant cells.

10. In the same vein, the experiment in Fig5 does not show that "Tubulin recruits PCNA to nuclear bodies" as stated in the title of this figure's legend. The experiment is only done in wildtype cells, so it can, at best, only demonstrate a correlation. To investigate the impact of g-tubulin on PCNA nuclear localization, the authors need to perform this experiment in g-tubulin-depleted cells, and in cells expressing the PIP mutant.

11. It is unclear why the authors measure "cell cycle chromobodies" to investigate cell cycle proliferation in Fig 7. This is a very esoteric approach. The flow cytometry approach in Fig 8a is much more powerful, and well established in the field. However, it is difficult from the quantification shown in Fig 8a to figure out if there are statistically significant differences between WT and PIP. The S and G2/M curves need to be separated on different graphs, and statistical analyses need to be performed.

12. While most of the work is cell cycle related, some of the experiments are done with DNA damage. It is unclear to me the relevance of these experiments. Is it implied that the same mechanism occurs after DNA damage treatment? If so, the localization, ChIP seq, and chromatin experiments with the PIP mutant need to be repeated with DNA damage, and drug sensitivity experiments with this mutant need to be included. But it seems to me that this would be a whole different project, so it may be better to simply remove the DNA damage experiments from this manuscript.

13. In general, I would refrain from using statements such as "Nothing is known about..." (page 6). In any case, it is unclear what it is meant in this particular circumstance.

Reviewer #3:

Remarks to the Author:

Authors have demonstrated a novel role for TUBG1 in aiding recruitment of PCNA to chromatin.

Authors show that TUBG1 is the major isoform enriched in chromatin and that TUBG1 co-fractionates with PCNA and is enriched in the nucleus at the same timeframe. Binding of TUBG1 and PCNA is also confirmed via IP. Authors show that γ -tubulin depletion reduces PCNA recruitment to the nucleus, however PCNA depletion does not affect γ -tubulin recruitment to the nucleus. ChIP seq analysis showed that PCNA and γ -tubulin bind chromatin at the same motifs, supporting the notion that TUBG1 assists recruitment of PCNA to chromatin.

Authors attempted to make TUBG1 that could not bind PCNA by mutating the PIP domain of TUBG1. It is unfortunate that disrupting the Q429, I432 and Y435 on TUBG1 was not sufficient to completely disrupt association with PCNA, as it would have been nice to properly assess dependence of PCNA on TUBG1 for recruitment to chromatin. Nonetheless the work presents an exciting novel mechanism to help explain the role of γ -tubulin in promoting cell proliferation.

Specific comments

Western blots shown in figure 3d are not very good quality due to very uneven loading and staining and are hence not very convincing, I suggest redoing these blots.

Diagram in figure 9c is very chaotic, I suggest simplifying this somehow to make it easier to digest, perhaps by adding a legend instead of labelling each individual protein.

LUND UNIVERSITY

Faculty of Medicine

Malmö 28-04-2021

Ph.D. Maria Alvarado Kristensson
Division for Molecular Pathology
SUS-Malmö, 59 Jan Waldenström street
Se-205 02 Malmö, Sweden
maria.alvarado-kristensson@med.lu.se

Point-by-point reply

Reviewers' comments:

Reviewer #1 (Remarks to the Author):

Corvaisier et al reports the interaction between gamma-tubulin and PCNA and its importance of chromatin loading of PCNA for proper cellular proliferation, especially in S phase and DNA damage response. Authors first isolated a proper antibody specifically detecting gamma-tubulin and used it for interaction study of PCNA. Authors rely on mainly cellular fractionation and microscopic observations to claim cytosolic PCNA is recruited to chromatin by gamma-tubulin. In addition, authors used chromatin immunoprecipitation and sequencing to show correlative accumulation of gamma-tubulin and PCNA in the genome. Lastly, authors analyzed pre-existing cancer patients data to support interdependent expression of gamma-tubulin and PCNA in various cancers. The manuscript contain somewhat surprising results and should be carefully examined whether all claims by authors are properly addressed in the manuscript. It will be important to address major as well as minor points listed below...

Major points.

Reviewer point #1

It is little striking observation that PCNA was observed mainly in cytosolic fraction in U2OS cells authors used. Although there are several papers describing cytosolic PCNA (eg. J.Exp.Med (2010) 207:2631) which describes PCNA movement to the cytosol during differentiation, majority of publications so far describe high enrichment of PCNA in the nucleus where PCNA functions.

Author response #1

Yes, indeed, there are multiple publications showing cytosolic PCNA. I would like to comment on the content of 5 of these publications. 1. doi:10.1016/j.febslet.2007.09.022. Here in figure 1a, in lane 2, there is a WB showing the cytosolic amount of PCNA in MCF10A. In this image, the authors compare various cell lines. Note that the amount of PCNA varies a lot between cell lines; 2. doi:10.1371/journal.pone.0117546. Here in fig. 3a, the authors show cytosolic PCNA in Sh-SY5Y cells; 3. doi: 10.1084/jem.20092241. Here in fig. 1b, the authors show cytosolic PCNA in neutrophils; 4. doi: 10.1016/j.febslet.2010.09.021. Here the authors analyze binding partners to cytosolic PCNA; 5. doi: 10.1038/srep35561. Here the authors study the function of cytosolic PCNA in acute myeloid leukemia. Based on the published data (especially in the article: doi:10.1016/j.febslet.2007.09.022), the amount of cytosolic PCNA depends on the cell type. Regarding our results, we have worked with different cell lines (MCF10A, U2OS, A549 and NHI3T3) and all of them show a considerable amount of PCNA in the cytosol. Note that our results are in line with the data presented in doi:10.1016/j.febslet.2007.09.022, that is,

LUND UNIVERSITY

Faculty of Medicine

Ph.D. Maria Alvarado Kristensson
Division for Molecular Pathology
SUS-Malmö, 59 Jan Waldenström street
Se-205 02 Malmö, Sweden
maria.alvarado-kristensson@med.lu.se

MCF10A has high amounts of cytosolic PCNA. We have added additional references to side 4, line 5 in the revised version of the manuscript. Furthermore, we have also been able to abolish the staining of PCNA with sg mediated knockdown of PCNA (Please see figure 3c). Moreover, both sh- and sg-mediated reduction of TUBG1 shift the cellular balance between cytosolic and chromatin-associated PCNA (Please see figure 3a, b, d and supplementary figure 4). Based on our observations and published data, I expect that the amount of cytosolic PCNA depends on the study cell type. Also, note that we analyzed all the fractions and that we do not discard any cell debris. Finally, I have noticed that in many studies, the authors used either total lysates in western blotting or chromatin fractions or both, but the cytosolic fraction is not specifically shown (as an example please see the following publication: doi:10.1016/j.molcel.2011.02.006). In this publication, the authors do not show the cytosolic levels of PCNA (please see figure 1a), but it is obvious that there is a difference between the amount of PCNA detected in the total lysate and in the chromatin, and that this difference may depend on the cytosolic pool of PCNA that is not analyzed by WB. Finally, independent of the amount of cytosolic PCNA, which based on published data is cell-type dependent, PCNA is translated in the cytosol and needs to be transported to the nuclear compartment and this needs to be regulated.

Reviewer point #2

ChIP Seq analysis should have valid negative control which do not show correlation with gamma-tubulin such as transcription factor. In addition, to confirm these colocalizations indeed reflect DNA replication fork, positive control (DNA replication enzyme such as polymerase, RFC, etc) should be included.

Author response #2

In the performed ChiP-seq experiments, we used in the ChiP-seq experiment using U2OS cells macrohistone 2A as a negative control and found no overlap between TUBG and macrohistone 2A. As it is difficult to predict the functions of TUBG in chromatin, in the ChiP-seq experiments using MCF10A cells, we used as a control a cell line that stably expressed TUBG-shRNAi, as the best control for the ChiP-seq of a protein is when reducing the amount of precipitated chromatin by decreasing the amount of the immunoprecipitated protein. In the latest performed ChiP-seq experiment using U2OS, we used both controls together, that is, U2OS cells stably expressing TUBG-shRNAi and, as suggested by you, ChiP-seq samples from a transcription factor. The aims of these last experiments were to distinguish between early, late, and dormant origins of replication in order to show to which origins TUBG was associated. Thus, we decided to immunoprecipitate with antibodies the following proteins: MCM5 (a marker for the active and dormant origin of replication), FoxM1 (transcription factor), TUBG1 and PCNA (a marker for the active origin of replication together with MCM5). Note that in the latest performed ChiP-seq, we have made 3 changes in the method used. The first change is the method used from fragmenting the DNA. Previously, we fragmented the DNA by sonication and now we used a micrococcal nuclease treatment for fragmenting the DNA. The second change is the use of a

LUND UNIVERSITY

Faculty of Medicine

Ph.D. Maria Alvarado Kristensson
Division for Molecular Pathology
SUS-Malmö, 59 Jan Waldenström street
Se-205 02 Malmö, Sweden
maria.alvarado-kristensson@med.lu.se

new monoclonal anti-PCNA antibody as the older polyclonal anti-PCNA antibody was discontinued. The third change is the use of Dynabeads for precipitation of the antibodies instead of using agarose coupled protein A. You can find the results of the new experiment containing 64 samples in figure 4e and supplementary Table 1. However, as there is an enrichment of firing origins close to transcription start sites, and TUBG1 regulates the transcriptional activities of E2Fs, together with the presence of TUBG1 in active origin of replications, it turned out that it was difficult to use a transcription factor as negative control. Note that apart of using U2OS cells, we also used U2OS cells stably expressing TUBG-shRNAi and found that the numbers of peaks retrieved in TUBG-shRNA-U2OS cells were reduced by approximately 50% in comparison with the number of peaks found associated with chromatin in U2OS cells, proving the affinity of the TUBG antibody and that the precipitated chromatin arise from TUBG immunoprecipitates. However, we performed the experiment as suggested, and the results are included in the revise version of the manuscript. Finally, despite the use of new anti-PCNA antibody, DNA-fragmentation and the pulldown method of the antibodies, we can draw the same conclusions as with the previous analysis, that is that PCNA and TUBG are together at active origins of replication.

Reviewer point #3

Authors showed gamma-tubulin deficiency showed S phase problem (Fig. 2). It could be due to other defects caused by gamma-tubulin deficiency. How to exclude other defects caused by gamma-tubulin deficiency?

Author response #3

Yes, we had the same issue and after giving it a thought, we decided that this issue was best answered by using different approaches. To avoid possible interference with cell cycle progression, the approaches were based on the sudden recruitment of PCNA caused by DNA damaging agents, such as NCS, in cells that: 1. had a reduced expression of TUBG (cells stably expressing TUBGshRNAi), 2. were expressing a sg-resistant TUBG1 (add-back experiment, i.e. cells stably co-expressing TUBG-sgRNAi and sg-resistant TUBG1) or 3. a sg-resistant TUBG1-PIP mutant (impairs the formation of the TUBG1-PCNA complex). We anticipated that treatment with NCS will cause the following: both (1) the reduced ability of the TUBG1-PIP mutant to form the TUBG1-PCNA complex and (2) the cellular reduction of TUBG levels will delay the recruitment of PCNA to chromatin, (3) in the presence of the TUBG1-wild type, the recruitment of TUBG1 and PCNA will be simultaneous (reinstated), leading to the recruitment of a larger pool of PCNA to chromatin. Indeed, we have been able to show that in cells with lower expression levels of TUBG1, treatment with NCS affected the recruitment of PCNA to chromatin (please see figure 3d and supplementary figure 4) and finally, treatment with NCS recruited the TUBG-PIP mutant (carries mutations in the PCNA-interacting domain) but the accumulation of PCNA was affected (please see figure 6e) and this effect was reversed in cells expressing TUBG1 (please see figure 6e). I would like to add that TUBG1 is a protein that it is difficult to work with. It takes at least 10 days to reduce the levels of TUBG below

LUND UNIVERSITY

Faculty of Medicine

Ph.D. Maria Alvarado Kristensson
Division for Molecular Pathology
SUS-Malmö, 59 Jan Waldenström street
Se-205 02 Malmö, Sweden
maria.alvarado-kristensson@med.lu.se

50%. During this process, the proliferation rate of the cell population is affected. Cells expressing shTUBG are able to establish a stable cell line with a 50% lower expression of TUBG, but it is different for cells expressing sg-TUBG1, as Cas9-sg knock-outs the activity of the TUBG1 gene. After 10 days, there are several scenarios in cells expressing sg-TUBG1. 1. The non-transfected cells grow faster than the GFP-crispCas9 positive cells and thus takes over the green cells. 2. Many cells have reached the point of TUBG concentration below 50% and thus have undergone apoptosis during the 10 days. 3. Despite carrying the Cas9-GFP-sgTUBG plasmid, the cell has been able to escape the actions of crispCas9 and thus the cell has a 100% TUBG protein levels. Based on all these events, in a sample containing cells from a good experiment, we might find 3 to 4 cells per sample that are green and containing reduced levels of TUBG. As it takes days to reduce the cellular levels of TUBG, we have tried to speed up this process by testing the auxin-degradon system (Doi: 10.1038/nmeth.1401). The auxin-degradon can induce degradation of a protein pool within one hour. Unfortunately, the aid tag, necessary for inducing the proteasomal mediated degradation of TUBG, is approximately 20 KDa, which results in an AID-TUBG protein with an expected molecular weight of approximately 70 KDa. When replacing the endogenous pool of TUBG with the aid-tagged TUBG, the resulting cell line shows a TUBG that is 50KDa and the addition of auxin does not lead to the degradation of the expressed TUBG, confirming that it is very difficult to reduce TUBG in a rapid way. The cells deplete TUBG in a rate that allows the cells to adjust to the new cellular levels of TUBG, and we think that once the cell cannot adjust to lower levels, the cells rapidly undergo apoptosis. Altogether, we thought that the best way of demonstrating that TUBG1 is necessary for the recruitment of PCNA was by being able to recruit TUBG1 to the nuclear compartment while PCNA remains in the cytosol. Note that cells co-expressing sg-TUBG1 and the TUBG1-PIP mutant are able to recruit TUBG1-PIP mutant to the chromatin but PCNA remains in the cytosol (please see figure 6e and 8b).

Reviewer point #4

In Figure 4, authors claimed enrichment of gamma-tubulin at the origin of replication. It will be important to know whether such enrichment are always observed or changed in time dependent manner. It will be quite important to understand dynamics of gamma-tubulin for active, dormant, late firing origins.

Author response #4

To understand the dynamics of TUBG1, we have both performed a new ChiP-seq experiment (enclosing 64 samples) and DNA fiber staining experiments to investigate the effect on dormant domain activation in U2OS cells co-expressing sgTUBG1 and TUBG1-PIP mutant. These data are presented in figure 4e, supplementary table 1 and figure 9. We find an enrichment of TUBG in early and late active origins is almost 100%. In contrast, the presence of TUBG was found only in approximately 50% of the dormant origins. This suggest that the recruitment of TUBG is before the firing of an origin. Finally, with the fiber assay, we found that the decreased recruitment of PCNA caused by the TUBG1-PIP-mutant leads to slow fork progression and

LUND UNIVERSITY

Faculty of Medicine

Ph.D. Maria Alvarado Kristensson
Division for Molecular Pathology
SUS-Malmö, 59 Jan Waldenström street
Se-205 02 Malmö, Sweden
maria.alvarado-kristensson@med.lu.se

shorter interorigin distance. Implying that dormant domains are fired but become stalled probably for the insufficient recruitment of PCNA. Please, note that these experiments are time consuming, extremely costly and very difficult to perform as all ChIP samples are connected to each other and for every ChIP-seq that did not work, all other ChIPs need to be redone.

Minor points.

Reviewer point #5

Overall, figures are too small to look through them and words in the figures are hard to read (resolution problem as well as size).

Author response #5

We have improved the resolution and the size of the text.

Reviewer point #6

In several places, “gamma-strings” are written as “gamma-Strings”. They should be consistently written.

Author response #6

We have controlled that “Strings” is only used at the beginning of a sentence, that is after a full stop of a previous sentence in the text.

Reviewer point #7

In Figure 1c, NSC and Neuron blot did not have any description what is the band means. Also, there appears to have several lanes, but not clear what each lane represents.

Author response #7

We apologize for the lack of information. We have added more information to figure 1 and rewrote the figure legend.

Reviewer point #8

In Figure 1a, arrows and arrowheads are used. In the legend, arrowheads and arrows are described as gamma-tubulin 1 and 2. But, sgRNA additions are also indicated as arrows. Authors need to make figure marks clearly to distinguish them.

Author response #8

We apologize for the mistake. We have changed the color of the sgRNA and sg-resistant protein-arrows. Blue arrows indicate now addition of either sgRNA and sg-resistant proteins.

Reviewer point #9

Figure 1C, the graph (top panel) labels are hard to follow.

Author response #9

We have improved the labeling of the figure.

Reviewer point #10

Page 7, authors described “nocodazole” was used for experiment of Figure 1d. However, in the figure, “Colcemid” is described. Should be consistent between figure and text.

Author response #10

We apologize for the mistake, which is mended in the revised version of the manuscript. Only colcemid was used in this study.

LUND UNIVERSITY

Faculty of Medicine

Ph.D. Maria Alvarado Kristensson
Division for Molecular Pathology
SUS-Malmö, 59 Jan Waldenström street
Se-205 02 Malmö, Sweden
maria.alvarado-kristensson@med.lu.se

Reviewer point #11

Figure 2a only shows one representing image. Quantification with more than 100 cells should be presented as graph.

Author response #11

We have added additional images to the figure 2a (please see supplementary figure 3a). Quantification with more than 100 cells are now included in supplementary figure 3b.

Reviewer point #12

Figure 3d, control 2 hour seems to have issue for loading. The representing blot should have all lanes properly showing necessary bands.

Author response #12

We have performed a new experiment and added it to the figure 3d.

Reviewer point #13

Page 11, "It should be noted that sg-mediated reduction of gamma-tubulin expression, also decreased the amount of gamma-tubulin associated with centrosomes, but ---"; Does this mean gamma-tubulin in centrosomes have retention function of proteins in centrosome?

Author response #13

No, we do not think so, but many scientists expect that a decreased expression of TUBG1 leads to the disassemble of the centrosome. However, other centrosomal markers do not disappear upon decreased levels of TUBG1 at the centrosome. In the revised version of the manuscript, we have measured the relative amount of TUBG1 associated with the pericentriolar matrix surrounding the centrosomes in U2OS and U2OS expressing TUBG1sgRNAi. As the nuclear- and PCM-associated pool of TUBG1 interacts with PCNA and with the growing PCNA foci, with this graph (please see supplementary figure 3b), we want to highlight that a sg-mediated decreased of TUBG1 levels affects the recruitment of PCNA to chromatin as well as decreased the amount of PCM-associated TUBG, which suggest that the nuclear- and PCM-associated TUBG pools are necessary for the transport and formation of PCNA foci in the chromatin.

Thank you for your helpful comments, time, and consideration.

Reviewer #2 (Remarks to the Author):

The authors convincingly demonstrate that PCNA and g-Tubulin co-localize, and directly interact through a PIP domain in g-tubulin. These are important results which validate the need for functional investigations into the roles of this interaction. To this end, the authors perform immunofluorescence and chromatin fractionation experiments, concluding that g-tubulin facilitates PCNA chromatin recruitment. Even though it is a very important part of the manuscript, I found this aspect of the manuscript to be much less convincing. Critical experiments to address the impact of the interaction are not presented, instead the authors rely on correlative results. The immunofluorescence experiments lack quantifications from multiple

LUND UNIVERSITY

Faculty of Medicine

Ph.D. Maria Alvarado Kristensson
Division for Molecular Pathology
SUS-Malmö, 59 Jan Waldenström street
Se-205 02 Malmö, Sweden
maria.alvarado-kristensson@med.lu.se

replicates and are thus difficult to interpret. The experiments are not performed with multiple knockdown cell lines to rule out off-target effects.

Specific comments:

Reviewer point #1

1. In Fig 2b, the quantification does not really match the blot. The upwards trend is clear on the blot, but not so much on the quantification.

Author response #1

We have added two additional experiments for obtaining a more accurate picture of the dynamics of the accumulation of TUBG1. Please, see figure 2b.

Reviewer point #2

2. A western blot showing the sgTub knockdown for the experiment in Fig3a is needed.

Author response #2

We cannot obtain the suggested western blot, as it is impossible to obtain a stable cell line of U2OS expressing sgTUBG1, as U2OS express only TUBG1 and depletion of the protein is lethal for U2OS cells. Cellular knockdown of TUBG1 with sgRNA is cytotoxic and thus, we can only study its effect by microscopy at a single-cell level. Depletion of TUBG below 50 % is cytotoxic. I would like to add that TUBG1 is a protein that it is difficult to work with. It takes at least 10 days to reduce the levels of TUBG below 50%. Cells expressing TUBGshRNAi are able to establish a stable cell line with an approximately 50% lower expression of TUBG, but it is different for cells expressing sg-TUBG1, as Cas9-sg knock-outs the activity of the TUBG1 gene. After 10 days expression of the Cas9-sg-TUBG construct, there are several scenarios. 1. The non-transfected cells grow faster than the GFP-crispCas9 positive cells and thus takes over the green cells. 2. Many cells have reached the point of TUBG concentration below 50% and thus have undergone apoptosis during the preceding 10 days. 3. Despite carrying the sgTUBG plasmid, the cell has been able to escape the actions of crispCas9 and thus the cell has a 100% TUBG protein levels. Based on all these events, in a sample containing cells from a good experiment, we might find 3 to 4 cells per sample that are green and containing reduced levels of TUBG. Consequently, it is impossible to analyze sg-TUBG1 expressing cells by western blotting, as it takes days to reduce the cellular levels of TUBG and at that time, the non-transfected cells are more in number than the sg-TUBG1 expressing cells. In an attempt to speed up this process, we tested to coexpress the auxin-degradon system (Doi: 10.1038/nmeth.1401) with sg-TUBG1. The auxin-degradon can induce degradation of a protein pool within one hour. Unfortunately, the aid tag necessary for the induction of the proteasomal mediated degradation of TUBG is approximately 20 KDa, resulting in an AID-TUBG protein with an expected molecular weight of approximately 70 KDa. When co-expressing sg-TUBG and aid-tagged TUBG, for replacing the endogenous pool of TUBG with the aid-tagged TUBG, in U2OS, MCF10A and A459 cells, the resulting cell lines carried a TUBG that is 50KDa and the addition of auxin did not lead to the degradation of the expressed

LUND UNIVERSITY

Faculty of Medicine

Ph.D. Maria Alvarado Kristensson
Division for Molecular Pathology
SUS-Malmö, 59 Jan Waldenström street
Se-205 02 Malmö, Sweden
maria.alvarado-kristensson@med.lu.se

TUBG – so it is very difficult to reduce TUBG in a rapid way! The cells deplete TUBG in a rate that allows the cells to adjust to the new cellular levels of TUBG and we think that once the cell cannot adjust to lower levels, due to the reduced TUBG protein levels in the mitochondria, the cells rapidly undergo apoptosis (DOI: 10.1038/s42003-018-0037-3).

Reviewer point #3

3. The authors used a single guide sequence and a single shRNA sequence throughout the manuscript. Some of the experiments are done with the guide, some are done with the shRNA, but no experiment is done with both of them to rule out off-target effects. To this end, they should use two independent sequences for each the guide and the shRNA.

Author response #3

We used both methods (Cas9-sg-TUBG and TUBG-shRNAi) to illustrate that independent of the chosen method or the sequence targeted, the recruitment of PCNA was impaired. The Cas9-sg-TUBG targets a sequence in the TUBG gene located in the N-terminal region (nucleotides 22-42), whereas TUBG-shRNAi targets a sequence at the C-terminal region of TUBG (nucleotides 1220-1238). Altogether, we are targeting 2 different sequences within the TUBG genes for silencing the expression of TUBG. Notably, both methods affect the recruitment of PCNA to chromatin (as example see: figure 3a,b [sg] and figure 3d, 4b,d,e and 6a [sh]. Examples of addback experiments are found in figure 6d,e [sg] and 4d, 6a [sh]). However, apart from confirming the specificity of sg-TUBG and TUBG-shRNAi in this manuscript, the specificity of both sg- and sh-TUBG constructs has also been previously proven by using cell lines that stably co-express either sg- or sh-TUBG1 and the sg- or sh-resistant TUBG1 genes, respectively (please see the following publications (DOI) : 10.1038/ncb1921, 10.1096/fj.11-187484, 10.1016/j.bbamcr.2017.10.008, 10.1016/j.heliyon.2016.e00166, 10.1038/s42003-018-0037-3, 10.1158/1541-7786.MCR-15-0063-T). Note that an additional control to cells co-expressing sg-TUBG1 and the sg-resistant TUBG1 are cells co-expressing sg-TUBG1 and the TUBG1-PIP mutant. These experiments are presented in figure 6d,e and 7–9, and supplementary figure 8b, and 9–11. We have also added to figure 3a and supplementary figure 3a the data showing the amount of PCNA and TUBG in chromatin in U2OS cells co-expressing sgTUBG1-TUBG1sg-resistant. Regarding the stable cell line expressing shTUBG, the add-back experiment, that is stable shTUBG cells coexpressing a GFP-tagged shTUBG resistant TUBG1 gene, is presented in figure 4d, 5b,c, and 6a.

Reviewer point #4

4. The authors need to include some type of quantification, from multiple replicates, for the immunofluorescence experiments in Fig 3b, c and Fig 5. In the absence of this, it is very difficult to assess if the authors' conclusions are supported by the data.

Author response #4

We have added the requested quantification from multiple replicates to figures 3b and c and supplementary figure 3b.

Reviewer point #5

LUND UNIVERSITY

Faculty of Medicine

Ph.D. Maria Alvarado Kristensson
Division for Molecular Pathology
SUS-Malmö, 59 Jan Waldenström street
Se-205 02 Malmö, Sweden
maria.alvarado-kristensson@med.lu.se

5. Fig 3d should include the whole cell extracts as control.

Author response #5

We have added an additional experiment with the requested controls. Please, see figure 3d in the revised version of the manuscript.

Reviewer point #6

6. Just from the chromatin fractionation shown in Fig 3d, it is difficult to conclude that PCNA chromatin localization depends on g-tubulin. This appears to be the main take home message of the manuscript, but it is not strongly supported by the data presented. The quantification shown in this figure shows a minimal difference, which is unlikely to be statistically significant. The authors should employ immunofluorescence experiments (with quantifications from multiple replicates) to investigate PCNA chromatin recruitment in shTUBG cells.

Author response #6

The data presented in figure 3d are statistically significant. By using a student t-test there is a significant difference in the recruitment of PCNA at 30 minutes ($P < 0.05$). A two-way ANOVA test was also used to evaluate the difference between U2OS and U2OS cells stably expressing shTUBG and the test shows also a statistical difference (** $P < 0.01$). However, as suggested we have also measured the recruitment of PCNA upon NCS treatment by immunofluorescence (please, see supplementary figure 4).*

Reviewer point #7

7. Page 11: it is confusing why the authors speculate that PCNA-tubulin complexes may limit the transport of PCNA to the nuclear compartment. Based on what do they speculate this? If anything, they show that the complexes promote PCNA chromatin binding, which would be the opposite.

Author response #7

We observed that most of the pool of the TUBG-PCNA complexes are located in the cytoplasm and a smaller pool of the complexes associated with chromatin, we speculated that the interaction of PCNA with TUBG is what keeps the protein either in the cytoplasm or the chromatin. So, in G1, little TUBG is found in the chromatin and consequently, PCNA is in the cytoplasm. Once the cell enters S-phase, TUBG accumulates in the chromatin, which also mediates the accumulation of PCNA to that location. We have explained this issue better in the new version of the manuscript (please, see page 11, first paragraph).

Reviewer point #8

8. Fig 4a should include all the time points shown for gTub, also for PCNA and H2A. Moreover, multiple replicates of this experiment should be performed. Finally, to rule out off target effects, it is critical that the experiments in Fig 4 are performed with an independent shRNA knockdown sequence, or with the sg cell line. It is difficult to make any conclusions in the absence of this.

Author response #8

LUND UNIVERSITY

Faculty of Medicine

Ph.D. Maria Alvarado Kristensson
Division for Molecular Pathology
SUS-Malmö, 59 Jan Waldenström street
Se-205 02 Malmö, Sweden
maria.alvarado-kristensson@med.lu.se

The suggested experiment is a huge experiment that it is very difficult to perform for the resources that my group have. So, we have tried to perform your suggestions according to our capacity. To understand the dynamics of TUBG1, we have both performed a new ChIP-seq experiment using U2OS and U2OS stably expressing shTUBG (in total 64 samples), and DNA fiber staining experiments using U2OS cells stably co-expressing sgTUBG1 and TUBG1-PIP mutant, to investigate the effect of TUBG on early, late and dormant activation of origin of replication (figure 4e, 9 and supplementary Table 1). In these experiments, we used the transcription factor FoxM1, as a control, as this control was suggested by one of the other reviewers that evaluated our manuscript. In addition, as we feel that the best control for the ChIP-seq of a protein is when reducing the amount of precipitated chromatin by decreasing the amount of the immunoprecipitated protein, we have also used a U2OS cells stably expressing TUBG-shRNAi. Note that in the latest ChIP-seq performed we have made 3 changes in the method used. The first change is the method used from fragmenting the DNA. Previously, we fragmented the DNA by sonication and now we used instead micrococcal nuclease treatment for fragmenting the DNA. The second change is the use of a new monoclonal anti-PCNA antibody as the older polyclonal anti-PCNA antibody is discontinuous. The third change is the use of Dynabeads for precipitation of the antibodies instead of using agarose coupled protein A. You can find the results of the new experiment in figure 4e and supplementary Table 1. The results show that it is difficult to use a transcription factor as negative control, as there is an enrichment of firing origins close to transcription start sites, and TUBG1 regulates the transcriptional activities of E2Fs, and it is present in active origin of replications. But, note that the number of peaks retrieved in TUBG-shRNA-U2OS cells were reduced by approximately 50% in comparison with the number of peaks found associated with chromatin in U2OS cells, proving the affinity of the TUBG antibody and that the precipitated chromatin arise from TUBG immunoprecipitates. Despite the changes of antibody, DNA-fragmentation and the pulldown method of the antibodies, we can draw the same conclusions as with the previous analysis, that is that PCNA and TUBG are together at active origins of replication.

Reviewer point #9

9. The CHIP-seq results (Fig4) do not show that “g-tubulin may facilitate the accumulation of PCNA in the nucleus”, as it is wrongly stated on page 17. Instead, these results show that g-tubulin binds to the same DNA regions as PCNA is known to bind to. This would in fact imply the opposite, that PCNA recruits g-tubulin to those regions. It is important that the authors address this issue by investigating the ChIP-seq localization of PCNA to PURA sites upon g-tubulin depletion, and in the PIP mutant cells.

Author response #9

Your assumption is based on a possible ability of PCNA to find its way to the PURA sequences, but, if PCNA requires assistance to find the PURA sequence, then TUBG needs to be recruited first to those regions. In line with an ability of TUBG to be first recruited to chromatin, we find that depletion of PCNA (sgPCNA expressing cells) does not impeded the recruitment of TUBG

LUND UNIVERSITY

Faculty of Medicine

Ph.D. Maria Alvarado Kristensson
Division for Molecular Pathology
SUS-Malmö, 59 Jan Waldenström street
Se-205 02 Malmö, Sweden
maria.alvarado-kristensson@med.lu.se

to chromatin. In contrast, depletion of TUBG reduced the chromatin associated pool of PCNA (Figure 3a). However, to further understand the function of TUBG in the activation of active, and early and late activation of origin of replication, we performed the ChiP-seq analysis by using the following antibodies: MCM5 (marker for active and dormant origin of replication), FoxM1 (transcription factor), TUBG1 and PCNA (marker for active origin of replication). The new ChiP-seq data (64 samples) show TUBG associated to almost all active origins of replication (96.8%) and to 51% of the dormant domains, suggesting that TUBG is recruited to a dormant domain before the activation of the origin and the subsequent recruitment of PCNA (please see figure 4e and supplementary figure 1). To finally study the consequences of disrupting the interaction between TUBG-PCNA complexes, we studied the effect of the TUBG1-PIP-mutant in DNA elongation and on firing of dormant origins by using a DNA fiber staining assay (Figure 9). We found that the decreased recruitment of PCNA caused by the TUBG1-PIP-mutant leads to slow fork progression and shorter interorigin distance.

Reviewer point #10

10. In the same vein, the experiment in Fig5 does not show that “Tubulin recruits PCNA to nuclear bodies” as stated in the title of this figure’s legend. The experiment is only done in wildtype cells, so it can, at best, only demonstrate a correlation. To investigate the impact of g-tubulin on PCNA nuclear localization, the authors need to perform this experiment in g-tubulin-depleted cells, and in cells expressing the PIP mutant.

Author response #10

We have rephrased the title to the figure 5 legend. We have tried to monitor live cells expressing sg-TUBG but the cells do not enter S-phase when coexpressing sg-TUBG and chromobody 10 days after transfection. The suggested experiment with the PIP mutant is presented in figure 7b. It has been impossible to obtain cells that co-express sg-TUBG and a GFP-tagged TUBG gene, as the resulting cell line expressed a non-tagged tubulin mutant.

Reviewer point #11

11. It is unclear why the authors measure “cell cycle chromobodies” to investigate cell cycle proliferation in Fig 7. This is a very esoteric approach. The flow cytometry approach in Fig 8a is much more powerful, and well established in the field. However, it is difficult from the quantification shown in Fig 8a to figure out if there are statistically significant differences between WT and PIP. The S and G2/M curves need to be separated on different graphs, and statistical analyses need to be performed.

Author response #11

We consider that TUBG forms a meshwork. The meshwork includes gamma-tubules, gamma-strings and the centrosome. Immunofluorescence of fixed cells gives only an image of the meshwork at the moment that the cell was fixed. No conclusions can be drawn on the dynamics of the meshwork. In the event that the immunostained cell population was synchronous in S-phase, we could place the observations from the fixed cells as occurring in S-phase, but, once again, we cannot predict or conclude anything about the dynamics of the meshwork during S-

LUND UNIVERSITY

Faculty of Medicine

Ph.D. Maria Alvarado Kristensson
Division for Molecular Pathology
SUS-Malmö, 59 Jan Waldenström street
Se-205 02 Malmö, Sweden
maria.alvarado-kristensson@med.lu.se

phase. Thus, the only possible way that we can think off monitoring the dynamics of the meshwork is by live imaging and this approach allowed us to record the movements of the centrosome, strings and tubules during S-phase progression. I do not agree that cell cycle chromobodies is an “esoteric approach”. I think that it is the only available way to relate the accumulation and location of PCNA with the dynamics of the meshwork during S-phase. We have separated the suggested graphs and present the statistical analysis (Figure 8a).

Reviewer point #12

12. While most of the work is cell cycle related, some of the experiments are done with DNA damage. It is unclear to me the relevance of these experiments. Is it implied that the same mechanism occurs after DNA damage treatment? If so, the localization, ChIP seq, and chromatin experiments with the PIP mutant need to be repeated with DNA damage, and drug sensitivity experiments with this mutant need to be included. But it seems to me that this would be a whole different project, so it may be better to simply remove the DNA damage experiments from this manuscript.

Author response #12

We were concerned that the effects observed in sh/sg-TUBG cells could depend on an effect on cell cycle progression caused by reduced levels of TUBG. Thus, we decided that this issue was best answered by using two different approaches. The first approach was to impair the formation of the TUBG1–PCNA complex. In this way, we anticipated that the TUBG1 mutant will be transported to the chromatin without PCNA. The second approach was to study the sudden recruitment of PCNA caused by DNA damaging agents, such as NCS, in cells that either had a reduced expression of TUBG or were expressing a sg-resistant TUBG-PIP mutant. Indeed, we have been able to show that in cells with lower expression levels of TUBG1, treatment with NCS affected the recruitment of PCNA to chromatin (please see figure 3d) and finally, treatment with NCS recruited the TUBG-PIP mutant (carries mutations in the PCNA-interacting domain) but the accumulation of PCNA was affected (please see figure 6e).

Reviewer point #13

13. In general, I would refrain from using statements such as “Nothing is known about...” (page 6). In any case, it is unclear what it is meant in this particular circumstance.

Author response #13

We have rephrased the sentence.

Thank you for your helpful comments, time, and consideration.

Reviewer #3 (Remarks to the Author):

Authors have demonstrated a novel role for TUBG1 in aiding recruitment of PCNA to chromatin.

LUND UNIVERSITY

Faculty of Medicine

Ph.D. Maria Alvarado Kristensson
Division for Molecular Pathology
SUS-Malmö, 59 Jan Waldenström street
Se-205 02 Malmö, Sweden
maria.alvarado-kristensson@med.lu.se

Authors show that TUBG1 is the major isoform enriched in chromatin and that TUBG1 co-fractionates with PCNA and is enriched in the nucleus at the same timeframe. Binding of TUBG1 and PCNA is also confirmed via IP. Authors show that γ -tubulin depletion reduces PCNA recruitment to the nucleus, however PCNA depletion does not affect γ -tubulin recruitment to the nucleus. ChIP seq analysis showed that PCNA and γ -tubulin bind chromatin at the same motifs, supporting the notion that TUBG1 assists recruitment of PCNA to chromatin.

Reviewer point #1

Authors attempted to make TUBG1 that could not bind PCNA by mutating the PIP domain of TUBG1. It is unfortunate that disrupting the Q429, I432 and Y435 on TUBG1 was not sufficient to completely disrupt association with PCNA, as it would have been nice to properly assess dependence of PCNA on TUBG1 for recruitment to chromatin. Nonetheless the work presents an exciting novel mechanism to help explain the role of γ -tubulin in promoting cell proliferation.

Author response #1

To further investigate the effect of TUBG and the TUBG1-PIP mutant on dormant activation of origin of replication, we have performed a DNA fiber staining experiment and added it to the figure 9. We found that a reduced loading of PCNA to fired dormant origins slow fork progression, and cells fire additional dormant replication origins to rescue replication, collectively causing a reduced inter-origin distance.

Specific comments

Reviewer point #2

Western blots shown in figure 3d are not very good quality due to very uneven loading and staining and are hence not very convincing, I suggest redoing these blots.

Author response #2

We have performed a new experiment and added it to the figure 3d.

Reviewer point #3

Diagram in figure 9c is very chaotic, I suggest simplifying this somehow to make it easier to digest, perhaps by adding a legend instead of labelling each individual protein.

Author response #3

We have modified figure 10c (previously figure 9c). I hope that the figure is now easier to understand.

Thank you for your helpful comments, time, and consideration.

Reviewers' Comments:

Reviewer #1:

Remarks to the Author:

Authors did all necessary experiments to strengthen the manuscript. Although it will be better to clearly see the PIP mutant showed clearer effect, due to technical difficulty of fully reduce the expression of TUBG, authors efforts addressed previous issues.

Reviewer #2:

Remarks to the Author:

The authors have satisfactorily addressed my comments. The revised manuscript is acceptable for publication in my view.

Reviewer #3:

Remarks to the Author:

Authors demonstrate a novel role for gamma-tubulin in recruiting PCNA to chromatin. Authors show that TUBG1 is the major isoform enriched in chromatin and that TUBG1 co-fractionates with PCNA and is enriched in the nucleus at the same timeframe. Binding of TUBG1 and PCNA is also confirmed via IP and could be largely perturbed by mutating a known PIPA PCNA binding motif on TUBG1. Authors show using quantification of immunofluorescence images that γ -tubulin depletion reduces PCNA recruitment to the nucleus, however PCNA depletion does not affect γ -tubulin recruitment to the nucleus. ChIP seq analysis showed that PCNA and γ -tubulin bind chromatin at the same motifs, and implicate γ -tubulin at active origins of replication. TUBG1 knock out rescued by PIPA TUBG1 mutants also showed slowed progression through S phase and G2/M and increased asymmetric replication forks, supporting the notion that γ -tubulin mediated PCNA recruitment helps facilitate DNA replication thereby cell division and proliferation. It is impressive and convincing that even incomplete perturbation of the interaction between γ -tubulin and PCNA had such a pronounced impact - importantly without any noticeable impact on microtubule nucleation in the cell.

I have only one comment, in Figure 9d the x axis legend suggests that TUBG1PIPA mutant rescues have greater than 100% asymmetric forks - I assume this was an error in the making of this figure.

LUND UNIVERSITY

Faculty of Medicine

Malmö 18-05-2021

Ph.D. Maria Alvarado Kristensson
Division for Molecular Pathology
SUS-Malmö, 59 Jan Waldenström street
Se-205 02 Malmö, Sweden
maria.alvarado-kristensson@med.lu.se

Point-by-point reply

REVIEWERS' COMMENTS:

Reviewer #1 (Remarks to the Author):

Authors did all necessary experiments to strengthen the manuscript. Although it will be better to clearly see the PIP mutant showed clearer effect, due to technical difficulty of fully reduce the expression of TUBG, authors efforts addressed previous issues.

Author response #1

Thank you for your helpful comments, time, and consideration.

Reviewer #2 (Remarks to the Author):

The authors have satisfactorily addressed my comments. The revised manuscript is acceptable for publication in my view.

Author response #1

Thank you for your helpful comments, time, and consideration.

Reviewer #3 (Remarks to the Author):

Authors demonstrate a novel role for gamma-tubulin in recruiting PCNA to chromatin. Authors show that TUBG1 is the major isoform enriched in chromatin and that TUBG1 co-fractionates with PCNA and is enriched in the nucleus at the same timeframe. Binding of TUBG1 and PCNA is also confirmed via IP and could be largely perturbed by mutating a known PIPA PCNA binding motif on TUBG1. Authors show using quantification of immunofluorescence images that y-tubulin depletion reduces PCNA recruitment to the nucleus, however PCNA depletion does not affect y-tubulin recruitment to the nucleus. ChIP seq analysis showed that PCNA and y-tubulin bind chromatin at the same motifs, and implicate y-tubulin at active origins of replication. TUBG1 knock out rescued by PIPA TUBG1 mutants also showed slowed progression through S phase and G2/M and increased asymmetric replication forks, supporting the notion that y-tubulin mediated PCNA recruitment helps facilitate DNA replication thereby cell division and proliferation. It is impressive and convincing that even incomplete perturbation of the interaction between y-tubulin and PCNA had such a pronounced impact - importantly without any noticeable impact on microtubule nucleation in the cell.

LUND UNIVERSITY

Faculty of Medicine

Ph.D. Maria Alvarado Kristensson
Division for Molecular Pathology
SUS-Malmö, 59 Jan Waldenström street
Se-205 02 Malmö, Sweden
maria.alvarado-kristensson@med.lu.se

I have only one comment, in Figure 9d the x axis legend suggests that TUBG1PIPA mutant rescues have greater than 100% asymmetric forks - I assume this was an error in the making of this figure.

Author response #1

Yes, indeed, this is a mistake. Thank you for your comment on figure 9d. We have mended the mistake in the revised version of the manuscript.

Thank you for your helpful comments, time, and consideration.